# Younger Dryas ice margin retreat triggered by ocean surface warming in central-eastern Baffin Bay

Mimmi Oksman[1,2], Kaarina Weckström[2,3], Arto Miettinen[4], Stephen Juggins[5], Dmitry V. Divine [4,6], Rebecca Jackson[7], Richard Telford [8], Niels J. Korsgaard [9] & Michal Kucera[7]

The transition from the last ice age to the present-day interglacial was interrupted by the Younger Dryas (YD) cold period. While many studies exist on this climate event, only few include high-resolution marine records that span the YD. In order to better understand the interactions between ocean, atmosphere and ice sheet stability during the YD, more high-resolution proxy records from the Arctic, located proximal to ice sheet outlet glaciers, are required. Here we present the first diatom-based high-resolution quantitative reconstruction of sea surface conditions from central-eastern Baffin Bay, covering the period 14.0–10.2 kyr BP. Our record reveals warmer sea surface conditions and strong interactions between the ocean and the West Greenland ice margin during the YD. These warmer conditions were caused by increased Atlantic-sourced water inflow combined with amplified seasonality. Our results emphasize the importance of the ocean for ice sheet stability under the current changing climate.

[1] Department of Geosciences and Geography, University of Helsinki, Gustaf Hällströmin katu 2a, Helsinki 00014, Finland. [2] Department of Glaciology and Climate, Geological Survey of Denmark and Greenland (GEUS), Øster Voldgade 10, Copenhagen 1350, Denmark. [3] Department of Environmental Sciences (ECRU), University of Helsinki, Viikinkaari 1, Helsinki 00014, Finland. [4] Norwegian Polar Institute, Fram Centre, Hjalmar Johansens gate 14, Tromsø 9296, Norway. [5] School of Geography, Politics and Sociology, Newcastle University, Newcastle upon Tyne NE1 7RU, UK. [6] Department of Mathematics and Statistics, University of Tromsø–The Arctic University of Norway, Hansine Hansens veg 54, N-9037 Tromsø, Norway. [7] MARUM-Center for Marine Environmental Sciences, University of Bremen, Leobener Strasse 8, Bremen 28359, Germany. [8] Department of Biology, University of Bergen, Postboks 7803, N-5020 Bergen, Norway. [9] Nordic Volcanological Center, Institute of Earth Sciences, University of Iceland, Sturlugata 7, IS-101 Reykjavík, Iceland. Correspondence and requests for materials should be addressed to M.O. (email: mimmi.oksman@helsinki.fi)

The Younger Dryas (YD) cold period at 12.9–11.7 kyr BP[1] interrupted deglacial warming in the Northern Hemisphere with a 10 °C drop in atmospheric temperatures[2]. Although the trigger of the event remains a matter of debate, it has been frequently associated with fresh water release into the North Atlantic[3, 4]. The resulting stratification would have weakened the Atlantic Meridional Overturning Circulation (AMOC), which is closely coupled to climate fluctuations[5], and disrupted heat transport to high latitudes[6]. However, a recent modeling study found that fresh water release alone could not have caused the cold period, but additional triggers were required[7]. The YD had a stronger impact on the Northern Hemisphere, although signs of this colder period have also been detected in tropical paleo-records[8].

Given the growing concern about recent changes in the Arctic region, including the polar amplification of atmospheric warming[9] and accelerated ice mass loss from the Greenland Ice Sheet (GrIS)[10], there is a need for high-resolution marine proxy data to better understand the interactions between the ocean and the outlet glaciers of the GrIS. So far, high-resolution studies on YD ocean surface conditions based on marine records from the Arctic region are rare[11, 12]. Baffin Bay (Fig. 1a) is a climatically sensitive region where sea ice cover prevails for most of the year, forming around NW Baffin Bay in September and reaching a complete sea ice cover by March[13], and where cold Arctic-sourced waters and warmer high-salinity Atlantic-sourced waters meet. The variability of these water masses has a direct impact on the marine-terminating portions of the ice sheets surrounding Baffin Bay. It is therefore a key region for investigating past climate oscillations and ocean-ice sheet interactions.

Modern Baffin Bay has a counter clockwise gyre circulation, consisting of the cold Baffin Current (BC) and the warmer West Greenland Current (WGC). The WGC consists of polar waters from the East Greenland Current (EGC) and of Atlantic-sourced waters from the Irminger Current (IC) (Fig. 1a). The EGC and IC water masses meet off southern Greenland and although becoming increasingly mixed as they travel northwards along the West Greenland coast, remain distinguishable[14]. The polar water component of the WGC loses its strength around the latitude of 64°N and does not penetrate into northern Baffin Bay, allowing Irminger Current water to rise toward the surface[15, 16]. The average modern July water temperature of the area is around 5 °C[16]. A CTD profile collected from the coring location on the Disko Shelf in September 2008 shows a relatively warm surface layer (ca. 4 °C at 4 m wd)[17] underlain by a layer of colder water down to ca. 100 m wd, below which temperature and salinity increase, yet remain slightly colder than the surface waters (Fig. 1).

During the LGM, ice margins from the northeastern Laurentide Ice Sheet (LIS), southern Innuitian Ice Sheet (IIS) and western Greenland Ice Sheet (GrIS) entered the Baffin Bay[18–20] and in the West Greenland sector ice reached the outer shelf[18]. Although the timing and the pattern of the deglaciation around Baffin Bay is poorly constrained, it has been suggested that the retreat of West Greenland ice streams was asynchronous[18, 21] and occurred between 17 and 11.5 kyr BP[22]. One of the most significant known retreats on the West Greenland margin was the collapse of the Jakobshavn Isbræ ice stream into Disko Bay during the YD[23]. Today the Jakobshavn Isbræ is one of the largest ice streams in Greenland, draining about 7% of the GrIS area[24]

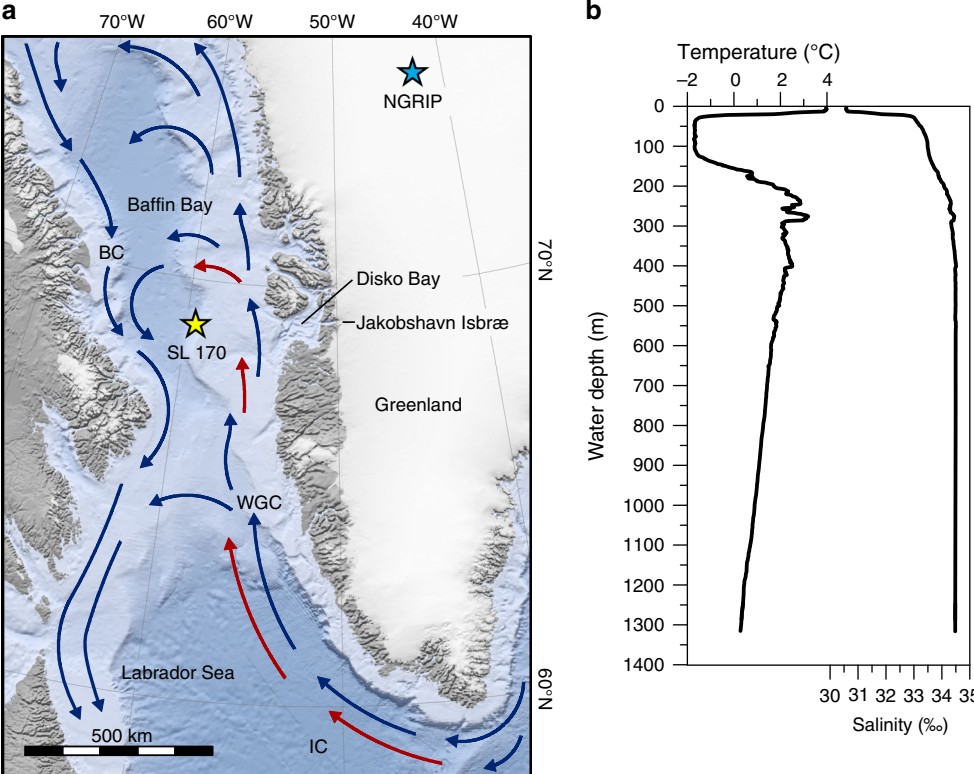

**Fig. 1** Map of the Baffin Bay showing the core location and CTD data from the study site. **a** Major surface currents in modern Baffin Bay; Baffin Current (BC), West Greenland Current (WGC), and Irminger Current (IC). Warm water currents are shown as red arrows and cold water currents as blue arrows. Coring site of SL 170 (marked with a yellow star) is located at 68° 58.15′ N 59° 23.58′ W. Location of the NGRIP is marked with a blue star and Disko Bay and Jakobshavn Isbræ are indicated on the map. Map data were retrieved from NSIDC[69] and NOAA NGDC[70]. **b** CTD cast made at the coring site during the cruise MSM09/2 in September 2008[17] shows water temperature (°C) and salinity (‰) profiles

**Table 1 Radiocarbon dates and calibrations from core SL 170**

| Sample ID | Material | Core depth (cm) | $^{14}$C age (yr BP) | Cal. yr BP | Calibrated age range (cal. yr BP), 95% confidence (2σ uncertainty) | Modeled age (cal. yr BP) |
|---|---|---|---|---|---|---|
| 55678.1.1 | Mixed benthic foraminifera | 24–27 | 9,668 ± 112 | 10,393 | 9,930–10,432 | 10,225 |
| 55679.1.1 | Mixed benthic foraminifera | 35–37 | 9,460 ± 80 | 10,145 | 10,094–10,500 | 10,309 |
| 55680.1.1 | Mixed benthic foraminifera | 55–57 | 9,833 ± 83 | 10,590 | 10,376–10,746 | 10,571 |
| 55681.1.1 | Planktic foraminifera (N. pachyderma) | 74–76 | 9,901 ± 82 | 10,676 | 10,651–10,966 | 10,796 |
| 55682.1.1 | Mixed benthic foraminifera | 74–76 | 10,028 ± 87 | 10,845 | | |
| 55682.2.1 | Mixed benthic foraminifera (duplicate) | 74–76 | 10,090 ± 97 | 10,915 | | |
| 55683.1.1 | Mixed benthic foraminifera | 98–100 | 10,243 ± 80 | 11,095 | 10,876–11,199 | 11,058 |
| 55683.2.1 | Mixed benthic foraminifera (duplicate) | 98–100 | 10,232 ± 137 | 11,069 | | |
| 55684.1.1* | Planktic foraminifera (N. pachyderma)* | 116–118 | 11,042 ± 107* | 12,387* | | |
| 55685.1.1 | Mixed benthic foraminifera | 116–118 | 10,274 ± 86 | 11,127 | 11,012–11,470 | 11,213 |
| Beta-344504* | Mollusc fragments* | 136–139 | 10,080 ± 50* | 10,909* | | |
| 55686.1.1* | Mixed benthic foraminifera* | 136–139 | 12,990 ± 117* | 14,592* | | |
| 58351.1.1 | Mollusc fragments | 159–160 | 10,755 ± 85 | 11,921 | 11,379–11,973 | 11,693 |
| 58352.1.1 | Mixed benthic foraminifera | 159–160 | 10,905 ± 85 | 12,219 | | |
| 58353.1.1 | Mixed benthic foraminifera | 180–181 | 10,671 ± 85 | 11,737 | 11,515–12,105 | 11,845 |
| 55687.1.1 | Mixed benthic foraminifera | 266–269 | 11,267 ± 100 | 12,642 | 12,250–12,656 | 12,498 |
| 58354.1.1 | Mixed benthic foraminifera | 288–290 | 11,150 ± 75 | 12,551 | 12,413–12,738 | 12,600 |
| 55688.1.1 | Planktic foraminifera (N. pachyderma) | 399–402 | 11,597 ± 104 | 12,933 | 13,040–13,368 | 13,213 |
| 55689.1.1 | Mixed benthic foraminifera | 399–402 | 11,944 ± 92 | 13,276 | | |
| KIA 40766 | Planktic foraminifera (N. pachyderma) | 484–488 | 12,730 ± 60 | 14,078 | 13,879–14,572 | 14,137 |
| 58355.1.1 | Planktic foraminifera (N. pachyderma) | 636–637 | 14,640 ± 130 | 17,137 | 16,434–17,462 | 16,991 |

All the radiocarbon dates were calibrated using the CALIB Rev 7.0.4 program[55] with the Marine13 calibration curve[56] and a ΔR of 140 ± 35 years. Gray dates marked with * are not included in the Bacon model. Mixed benthic foraminifera include the species *Cassadulina reniforme, Cassadulina neoteretis, Elphidium excavatum, Melonis barleeanus, Astrononion gallowayi,* and *Islandiella norcrossi*

and producing 10% of the total iceberg discharge from the GrIS[25]. It has been proposed that ice stream retreat in the east, south−east, south−west, and west of Greenland may have been triggered by the incursion of warmer waters during the YD[23, 26–29], and that the ice margin retreat was out of phase with atmospheric temperatures over Greenland. These fast-flowing ice streams are strongly influenced by sea surface temperatures (SSTs) at their front[30, 31] and contribute today one-third[32] of the yearly ice loss from the GrIS.

Here we use the high-resolution sediment core SL 170, strategically located close to the maximum post-LGM extent of the Jakobshavn Isbræ (Fig. 1), to investigate the interaction between ocean surface conditions and the West Greenland ice margin from 14.0 to 10.2 kyr BP, including the entire YD stadial. August sea surface temperatures (aSSTs) were quantitatively reconstructed from fossil diatom assemblages using a weighted-averaging partial least squares (WA-PLS) transfer function[33], which is based on a large northern North Atlantic calibration data set[34]. Q-mode factor analysis, using the same calibration data set, was also conducted to relate modern diatom assemblages to

different ocean water masses and applied downcore to assess changes in the relative influence of different water masses off West Greenland during the studied time period. The proportion of the bulk sediment in the clay and silt (<63 μm) size fraction was used as a proxy for the influence of the glacier meltwater suspension plume[35], as size fractions larger than this are too large to be carried in the plume[36]. This high-resolution reconstruction of sea surface conditions in Baffin Bay during the deglacial shows warmer SSTs during the YD, and indicates these played a key part in the Jakobshavn Isbræ ice margin retreat. We conclude that these warmer conditions were caused by an enhancement of Atlantic-sourced water inflow together with increased seasonality. Based on our data, the retreat of the Jakobshavn Isbræ during the cold YD occurred mainly by calving, while surface/subsurface melt dominated during a warmer climate before and after the YD.

## Results

**Chronology and sedimentation.** The chronology of the 683 cm long sediment core SL 170, representing ca. 17−10 kyr BP, is

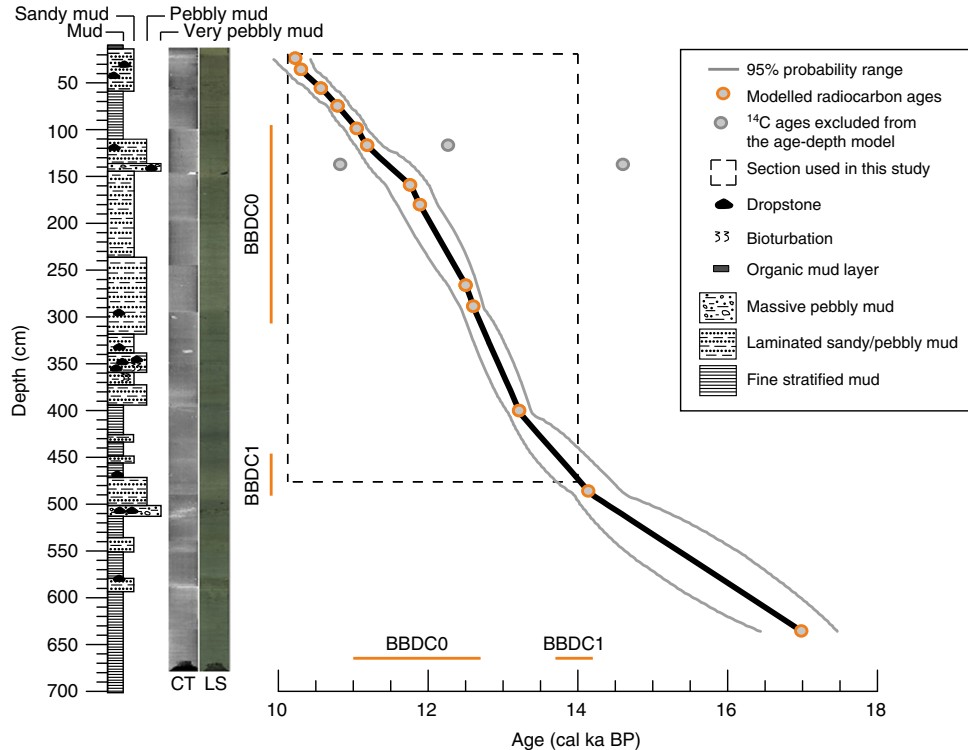

**Fig. 2** Age-depth model and lithostratigraphy of core SL 170. The age-depth model is based on 18 radiocarbon dates starting at 24 cm depth[37]. Dashed lines limit the studied interval from 14.0 to 10.2 kyr BP (24–475 cm). Dark gray line is the 95% uncertainty range. Simplified stratigraphy log, computerized tomography (CT) image and line scan (LS) high-resolution digital images are on the left. Orange lines show the Baffin Bay Detrital Carbonate Events, BBDC0 and BBDC1 in the core SL 170

based on 18 AMS[14]C dates (Table 1, Fig. 2) and has been previously presented by Jackson et al.[37]. The section (475–24 cm) used in this study represents the age range from ca. 14.0 to 10.2 kyr BP (Fig. 2), covering the end of the Bølling-Allerød, the YD and the onset of the Holocene. The sediment lithofacies shows two intervals of fine stratified muds from ca. 14.0 to 13.2 kyr BP (474–398 cm) and from ca. 11.1 to 10.6 kyr BP (110–58 cm) and deposition of coarse material including dropstones between these fine sediment layers (Fig. 2). A distinct peak of coarser sediment (ca. 68% >63 μm) is observed between 13.2 and 13 kyr BP. After this peak, the proportion of finer sediments (carried by the meltwater suspension plume) decreases continuously toward the top of the sediment record (Fig. 3e). Sedimentation rates are highest (avg. 168 cm/ka) between ca. 13.2 and 11.7 kyr BP (Fig. 3f) and the mean resolution of the studied section is 8.42 years cm$^{-1}$, allowing us to reconstruct sea surface conditions at an exceptionally high sub-decadal resolution.

**Surface conditions**. Diatom inferred aSSTs vary from 3.0 °C to 4.5 °C between ca. 14.0 and 10.2 kyr BP (Fig. 3a). Before the onset of the YD, aSSTs start to increase at ca. 13.4 kyr BP and the warming generally continues through the YD (12.9–11.7 kyr BP), but is punctuated by colder episodes (down to 3.5 °C) at around 12.6, 12.4, and 11.8 kyr BP. After the YD, the conditions are characterized by a notable gradual decrease in aSSTs (ca. 1 °C over 800 years), which continues to ca. 10.9 kyr BP when the lowest temperatures of 3.0 °C are recorded. After ca. 10.9 kyr BP temperatures rise rapidly by ca. 1.5 °C, staying at a higher although variable level throughout the rest of the record.

Q-mode factor analysis revealed a good correspondence between the modern distribution of the most common diatom assemblages (factors; Supplementary Note 1, Supplementary Fig. 1) in the North Atlantic calibration data set and the

downcore assemblages in the central-eastern Baffin Bay sediment core. The two main contributors to the SL 170 diatom assemblages are the same as in the Baffin Bay today (Supplementary Note 1, Supplementary Table 1): the Arctic Water assemblage and the Marginal Ice Zone (MIZ) assemblage. The Arctic Water assemblage is typically found between Polar and Atlantic waters (north of Iceland and in Baffin Bay), whereas the MIZ assemblage is indicative of a cold and fresh meltwater layer. At present this assemblage has its highest abundances along the spring sea ice limit in Fram Strait/NE Greenland and in the North Water Polynya (Supplementary Fig. 1). The downcore variability of the MIZ assemblage suggests two pronounced (ca. 14.0–13.4 and ca. 11.2–10.8 kyr BP) and two less pronounced (ca. 12.7–12.0 and ca. 11.7–11.3 kyr BP) melt periods (Fig. 3b), while the Arctic Water assemblage decreases in dominance over the same time periods (Fig. 3c). Both the Arctic Water and the MIZ assemblages show high variability between ca. 12.5 and 12.1 kyr BP. The factor for the East−West Greenland Current assemblage increased to slightly higher levels (from ca. 0.04 to 0.1) at ca. 13.4 kyr BP, also showed high variability between ca. 12.5 and 12.1 kyr BP, and a short episode of lower levels (down to 0.03) at ca. 11.2–10.9 kyr BP, before a continuous increase toward the beginning of the Holocene (Fig. 3d). The contribution of all other factors to the downcore assemblages are shown in Supplementary Fig. 2. The relative abundance of the most common diatom species and diatom concentrations in core SL 170 and the relative abundances of the same species in present-day Baffin Bay surface sediments are presented in Supplementary Note 2 and Supplementary Figs. 3, 4.

## Discussion

The post-LGM retreat of the West Greenland ice margin began in the Uummannaq Trough, north of Disko Bay, at ca. 15.0 kyr

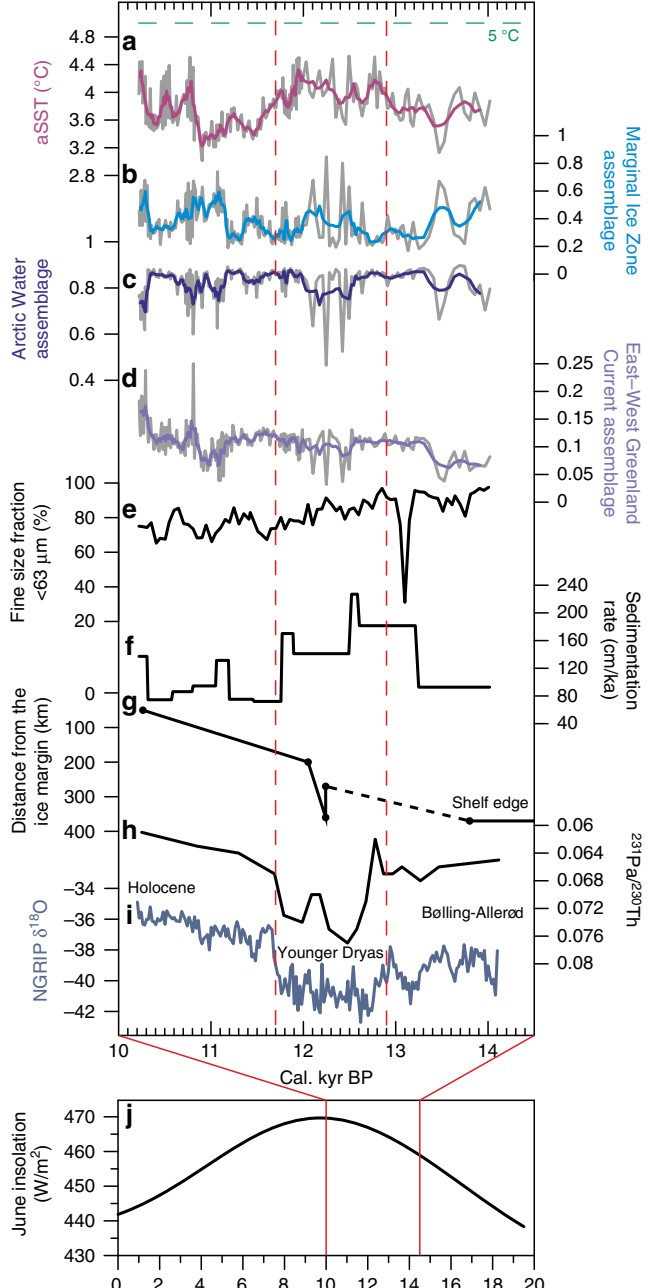

**Fig. 3** Results from core SL 170 compared with relevant paleoclimate data. **a** August sea surface temperature (aSST) reconstruction (°C). **b** Marginal Ice Zone assemblage factor. **c** Arctic Water assemblage factor. **d** East−West Greenland Current assemblage factor. Thick lines in **a**–**d** indicate smoothed records (five points weighted average). **e** Fine grain size fraction; <63 µm (%). **f** Sedimentation rates (cm/ka). **g** Time-distance diagram of Jakobshavn Isbræ extent modified from Cofaigh et al.[18]. Distances are calculated from the modern Greenland Ice Sheet margin. Black circles indicate [14]C dates, for the dashed line the ice margin position is unknown. **h** AMOC rate presented as sedimentary [231]Pa/[230] Th from the subtropical North Atlantic ocean[5]. **i** Stable oxygen isotope δ[18]O record from the NGRIP[1]. **j** Mean monthly insolation values for June at 68°N for the last 20 kyr[46]. Green dashed line indicates average modern summer (July) SST (5 °C)[16]. Red dashed lines marks the limits of YD (12.9–11.7 kyr BP) based on NGRIP[1]. When comparing with the previously published data included in the figure, we would like to refer the reader to our Supplementary Note 3, where the effect of different reservoir age corrections on our chronology has been tested

BP[38]. In the Disko Trough the retreat appears to have begun later, implying that the retreat of adjacent West Greenland outlet glaciers was asynchronous[18, 21]. The Uummannaq ice stream retreated during the Bølling-Allerød interstadial and later stabilized on the mid-shelf for the cold YD event. In contrast, the rapid retreat (collapse)—after a short-lived YD advance (Fig. 3g) —of the Jakobshavn Isbræ ice stream at ca. 12.2 kyr BP[18, 23, 39], in the middle of the cold YD event, remains puzzling. Asynchronous ice margin retreat has also been observed in SE Greenland, where it has been hypothesized to be controlled by the varying influence of IC due to shelf bathymetric constrictions[26].

Modern Jakobshavn Isbræ calving is known to be strongly affected by warmer Atlantic waters from the Irminger Sea[30]. Rinterknecht et al.[23] put forward the hypothesis that the Jakobshavn Isbræ retreat at ca. 12.2 kyr BP was triggered by warm water incursion via a strengthened WGC[23, 40]. They argue that combined with amplified SST and surface air temperature seasonality, linked to weak overturning during the reduced AMOC[5], the strengthened WGC could have increased the basal melt under the ice shelf[23]. Such a hypothesis is, however, difficult to reconcile with the YD Northern Hemispheric cooling and is until now not supported by marine records from West Greenland. Two recent studies[38, 41], based on dinoflagellate and foraminifera species data, point to the influence of warmer Atlantic water on the West Greenland shelf during the YD. However, Gibb et al.[41] do not discuss their species data further, and the lower resolution and coarse chronology in Sheldon et al.[38] means that the timing of the faunal changes are poorly constrained. Jennings et al.[39] on the other hand, suggested that the collapse must have occurred rapidly by calving under very cold conditions with quasi-perennial sea ice, since they found no evidence of warmer ocean currents or glacier melting. The alternative explanation for the retreat proposed by Cofaigh et al.[18]—that the rapid YD advance of the glacier was a surge (controlled only by ice thickness and subglacial bedrock topography), which would have thinned the outlet and made it vulnerable to collapse—would agree with the scenario proposed by Jennings et al.[39]. The exact behavior of the Jakobshavn Isbræ retreat after the collapse is unclear and two alternatives, either a sustained retreat[39] or a second collapse at ca. 10.2 kyr BP[23], have been proposed.

Our record from core SL 170 provides the first high-resolution reconstruction of sea surface conditions in Baffin Bay during the deglacial. The presented chronology for the core SL 170 (Fig. 2) is exceptional since radiocarbon chronologies from the deep marine environments are rare. Assessing the local [14]C reservoir age (ΔR) for radiocarbon chronologies is challenging due to its variation through time, thus we addressed this issue by testing a range of ΔR corrections (Supplementary Note 3) and created several age-depth models only varying the constant ΔR (0 ± 0–1000 ± 100 years) (Supplementary Fig. 5, Supplementary Table 2) and varying ΔR over time (200 ± 100, 400 ± 100 and 1000 ± 100 years for the YD and 140 ± 35 years for the early Holocene and for the Bølling-Allerød) (Supplementary Fig. 6). Based on this exercise, we conclude that (except for the extreme scenario of 1000 ± 100 years) the main findings of our study are not affected by the changing ΔR and the chosen ΔR of 140 ± 35 years used in previous studies[21, 38, 39] is justified. The radiocarbon chronology is notably strengthened by the correlation of the detrital carbonate rich layers in core SL 170[37] with known Baffin Bay Detrital Carbonate events (BBDC)[20] indicating meltwater and iceberg discharge in Baffin Bay and the Labrador Sea region. Due to its location and exceptionally high resolution the record facilitates direct comparison with the known behavior of Jakobshavn Isbræ[18, 21, 23, 39], and provides strong support for warmer sea surface conditions that is suggested as a principal driving mechanism in the hypothesis of Rinterknecht et al.[23].

The reconstructed aSSTs show an obvious mismatch with the $\delta^{18}O$ record from the North Greenland Ice Core Project (NGRIP)[1] (Fig. 3a, i) and with SST reconstructions from the northern North Atlantic[42, 43] which both, unlike the SL 170 record, point to a strong cooling over the YD period. This indicates that the oceanic regime in Baffin Bay was out of phase with the atmospheric forcing and North Atlantic surface ocean conditions during the YD.

Instead of recording a cooling of the ocean surface during the YD cold period, the reconstructed aSSTs show warmer conditions (compared to aSSTs before and after YD) starting at ca. 13.4 kyr BP and continuing to the end of the YD (Fig. 3a). The highest sedimentation rates, accompanied by lower abundances of MIZ diatoms, are observed during the YD (Fig. 3b, f), post-dating the onset of warmer conditions. This period of high sedimentary input correlates with the sequence of coarser sediment (pebbly mud, including dropstones) (Fig. 2) and a peak of coarser sediment at ca. 13.2–13.0 kyr BP (>63 μm; IRD, Fig. 3e), suggesting enhanced interaction between the ocean and the Jakobshavn Isbræ. Although our site is far offshore and hence also includes a proportion of IRD carried by icebergs in the WGC and potentially icebergs from northern Baffin Bay[37, 39], the bulk of IRD is typically lost relatively close to the source in particular under warmer ocean conditions[36, 44]. Our data indicate that warming ocean surface conditions led to intensive calving and retreat of the Jakobshavn Isbræ beginning already around 13.4 kyr BP. Sedimentation rates are markedly higher between ca. 13.2 and 11.7 kyr BP, while the decreasing amount of fine sediments from the meltwater suspension plume imply increasing distance from the retreating glacier[35]. The high variability in the Arctic Water and MIZ assemblages between 12.5 and 12.1 kyr BP indicates strongly fluctuating surface water conditions between fresh meltwater and Arctic water suggesting that calving and associated melting of icebergs occurred in distinct episodes.

Unlike Hogan et al.[21], who concluded that the retreat of Jakobshavn Isbræ was modulated only by topography and ice dynamics rather than climatic or oceanic drivers, we propose that both the YD re-advance and subsequent retreat were governed by a combination of an oceanic driver and topographic control[21]. Here, the advance would represent a surge leading to ice thinning, which in turn caused the rapid retreat (collapse) documented in previous studies[18, 21, 23, 39]. The uplift adjustment used in Rinterknecht et al.[23] did not take into account the gravitational attraction of water toward ice (see Ullman et al.[45]). If this is accounted for, the collapse would have occurred at the end of the YD, when we see a marked drop in sedimentation rates indicating that the glacier had retreated away from the outer shelf (Fig. 3g)[21]. The advance and collapse of Jakobshavn Isbræ are preceded by the marked calving episodes between 12.5 and 12.1 kyr BP as evidenced by the diatom data and supported by an increased amount of dropstones during this interval.

The increase in the East–West Greenland component at ca. 13.4 kyr BP in our record (Fig. 3d) indicates an enhanced WGC, which could plausibly be linked to the suggested intensification of the IC during the Allerød and the YD[27, 29]. Intensified IC has been proposed to have caused ice retreat in SE and SW Greenland[26–28]. This warmer water inflow, and the increasing solar insolation in the Northern Hemisphere[46] (Fig. 3j)—that has been shown to drive substantial warming of surface waters[47], are likely responsible for the increased SSTs in Baffin Bay. As our reconstructed SST increase precedes the AMOC reduction by several centuries (even allowing for dating uncertainties at this time, Figs. 2, 3a, h), we suggest that the weakening overturning circulation did not play a key role in the initial increase in SSTs and seasonality (as suggested by Rinterknecht et al.[23]). Instead, increased seasonality, as indicated by our reconstructed higher

summer temperatures, with open water conditions (based on diatom concentrations, Supplementary Fig. 3) during the summer and extensive sea ice cover during the winter[39, 41], was a result of increasing summer insolation (Fig. 3j). This would agree with earlier terrestrial studies from southern Greenland and Baffin Island suggesting amplified seasonality with warmer summers and colder winters[48–50] and from central East Greenland where warmer summer temperatures are even suggested to trigger ice margin retreat during the YD[51].

The ocean cooled by ca. 1 °C over the 800 years following the end of the YD (Fig. 3a). This is likely a product of increased surface melt (and calving) from glaciers surrounding Baffin Bay. The influence of the WGC appears to weaken and the Arctic water influence is stronger (Fig. 3c, d) during the coldest interval of our record (~10.9 kyr BP), before a clear increase in the East–West Greenland Current influence and an aSST increase in the early Holocene. MIZ diatoms indicate a period of exceptionally pronounced melt from 11.2 to 10.8 kyr BP (Fig. 3b). This coincides with increased sedimentation of fine stratified mud (Fig. 2) and may indicate a phase of intensive retreat caused predominantly by surface melt of the Jakobshavn Isbræ. It appears therefore that there were two main modes of Jakobshavn Isbræ retreat; via surface melting during warm climate periods and by calving, triggered by the warmer ocean, during the low atmospheric temperatures of the YD. This agrees well with another sediment study from the outer Disko Trough[39] in which 'calving retreat sediments' were recorded until 11.4 kyr BP, and between 11.4–10.5 kyr BP the sediment contained glacial marine sediments with a greater component of fine debris originating from the Jakobshavn Isbræ and northern Baffin Bay. Although Jakobshavn Isbræ appears to have retreated steadily from the outer shelf after its collapse, based on the decreasing amount of fine sediments from the meltwater suspension plume[35] (Fig. 3e), the coarser sediment section, including high IRD and dropstones and increased sedimentation rates after ca. 10.5 kyr BP, could possibly indicate another rapid retreat or a collapse.

Although several studies from the northern Labrador Sea and the North Atlantic have reported a climate shift from glacial to deglacial conditions around ca. 12.2 kyr BP during the mid-YD[11, 40, 42, 52], we find no evidence of a similar shift occurring in central-eastern Baffin Bay. Based on sediment stratigraphies off Newfoundland, Pearce et al.[11] suggested that the end of the YD in the area was initiated by changes in the oceanic regime, e.g., a weakening of the Labrador Current and a simultaneous northward shift of the Gulf Stream. While we see no evidence of increased Atlantic water influence at the end of the YD, our results (Fig. 3a, d) show enhancement of the WGC at ca. 10.9 kyr BP, including a pronounced peak at ca 10.8 kyr BP which is exceptionally high for this assemblage in the area compared to average modern conditions (Supplementary Table 1). This agrees with a previous study inferring warmer waters entering the northern Baffin Bay at the time[53] with increased summer air temperatures on Baffin Island[54] and with the rapid deglaciation of the Sermilik Fjord in SE Greenland[26].

The collapse of the Jakobshavn Isbræ on the West Greenland coast during the YD has to date been an enigma, because it occurred during a period of cold climate over Greenland. Our high-resolution reconstructions of ocean surface conditions clearly point toward a warmer ocean during the YD playing a key part in the Jakobshavn Isbræ ice margin retreat alongside topographic controls and ice sheet dynamics[21]. We conclude that these warmer conditions were caused by increased solar forcing together with an enhancement of Atlantic-sourced water inflow. We observe increased seasonality with longer ice-free periods during the summer than earlier suggested[39]. Based on our sedimentological and diatom data, the retreat of the Jakobshavn

Isbræ during the cold YD occurred mainly by calving, while surface/subsurface melt dominated during the warm conditions at the end of the Bølling-Allerød and at the onset of the Holocene. Our results show that the ocean had significant interactions with the Greenland Ice Sheet in Baffin Bay during the YD, and emphasize the importance of the ocean for ice sheet dynamics under the ongoing climate warming.

## Methods

**Coring and sampling**. The marine sediment core SL 170 was recovered from the deepest ploughmark on the Disko Trough outer shelf in central-eastern Baffin Bay (68° 58.15′ N 59° 23.58′ W; Fig. 1). The 683 cm long sediment core was collected from 1078 m water depth using a gravity corer (Schwerelote) from the research vessel *Maria S. Merian* during the cruise MSM09/2 in September 2008. The core was split into an archive and a working half and both halves were stored at < 4 °C. In total, 228 diatom samples were taken from the sediment core at 1 cm intervals from the core depths 24–195 cm (ca. 11.9–10.2 kyr BP), and at 5 cm intervals from the depths of 195–475 cm (ca. 14.0–11.9 kyr BP). Grain size distribution was measured from 77 samples at ca. 50 years resolution using a Malvern Mastersizer 2000(G) particle size analyser.

**Chronology**. Twenty-one Accelerated Mass Spectrometry (AMS) [14]C measurements from the core SL 170 were made at the Laboratory for Ion Beam Physics, ETH Zurich (Table 1). The radiocarbon samples consisted of mixed benthic foraminifera, or planktic foraminifera (*Neogloboquadrina pachyderma* sinistral, left coiling) picked from the >150 μm fraction, and two mollusc fragments were also dated (Table 1). The benthic foraminifera assemblages used for radiocarbon dating included the species *Cassadulina reniforme*, *Cassadulina neoteretis*, *Elphidium excavatum*, *Melonis barleeanus*, *Astrononion gallowayi*, and *Islandiella norcrossi*. Due to the varying contributions of these species to the assemblages downcore, mixed samples were required to provide enough carbonate for radiocarbon dating. Benthic assemblage analysis (Jackson, R. personal communication) indicate that miliolid spp. were scarce downcore. On the rare occasion when miliolid spp. (which are known for giving too old ages) were found, they were not included in the samples sent for radiocarbon dating.

The radiocarbon ages were calibrated in the CALIB Rev 7.0.4 program[55] using the Marine13 calibration curve[56] and a local reservoir correction (ΔR) of 140 ± 35 years[57] as has been used in previous studies in Disko Bay e.g., refs [21],[38],[39]. Bacon software[58] was applied to create an age-depth model (Fig. 2). We find the ΔR value of 140 ± 35 years justified after running several age-depth models with varying the ΔR value and by correlating the timing of detrital carbonate layers in the sediment[37] with the timing of the known Baffin Bay Detrital Carbonate events BBDC0[20],[39] and BBDC1[20] (Supplementary Note 3). Three radiocarbon ages were considered outliers since they were clearly outside of the 95% uncertainty range (Fig. 2) and therefore were excluded from the age-depth model. Due to an absence of biogenic carbonate, the top 24 cm of the core could not be dated and was therefore not included in the age-depth model. Sedimentation rates were calculated between radiocarbon-dated intervals using the Bacon output file of age vs. depth.

**Diatoms and aSST transfer function**. Diatom samples were prepared using hydrogen chloride (HCl), hydrogen peroxide ($H_2O_2$), and clay separation[59]. A minimum of 300 diatom valves were identified at each level using ×1000 magnification[60]. *Chaetoceros* resting spores are routinely excluded from the total diatom counts because they generally completely dominate fossil assemblages yet they show negligible sensitivity to SST[61]. aSST was reconstructed using a weighted-averaging partial least squares (WA-PLS) transfer function[33] based on a modern calibration data set[34] consisting of 183 surface sediment samples from the North Atlantic, including the Labrador Sea, the Nordic Seas, and Baffin Bay. The 2-component WA-PLS model has a root mean square error (RMSE) of 1.14 °C, $r^2$ (the coefficient of determination between observed and inferred values) of 0.92 and a maximum bias of 2.81 °C. The prediction error was estimated using $h$-block cross validation[62] in which samples closer than a cutoff distance ($h$) from a target sample were excluded from contributing to the prediction of that sample. $h$-block cross validation was selected to allow for spatial dependency in the calibration data, which can lead to underestimation of the prediction error because of pseudoreplication. The cutoff distance ($h$) was estimated by assessing the spatial structure of the residuals of the surface sample predictions. Specifically, $h$ was estimated as 495 km using the range of a circular variogram fitted to the detrended residuals[63]. The optimal number of WA-PLS components was determined using a randomization $t$-test[62] applied to $h$-block prediction errors.

**Factor analysis**. A Q-mode factor analysis[64] applied to the modern diatom calibration data set[34] revealed eight factors (i.e., most common compositions of diatom assemblages) that are related to distinct ocean water masses; Marginal Ice Zone, Arctic Waters, Greenland Arctic Waters, East–West Greenland Current, Transitional Waters, Sub-Arctic Waters, North Atlantic Current, and Norwegian

Atlantic Current (Supplementary Fig. 1) accounting for 95% of the total variance in the data set. This analysis is an extension of earlier studies[65],[66], which use the original version of the method presented by Imbrie and Kipp[64]. The method is essentially a dimension reduction technique, which describes the variability within the set of observed and correlated variables in terms of a lower number of unobserved variables called factors. When applied to microfossil data from marine sediments, the Q-mode factor analysis effectively groups the original taxa into a small number of factors or "assemblages" that are indicative of specific environmental conditions (in this case relating to distinct ocean water masses). The factor model can then be applied downcore to reconstruct changes in the relative importance of the various assemblages (and hence water masses) through time. More specifically, given the $M$ by $N$ row-normalized matrix $U_{cd}$ constructed of $m = \{1,…, M\}$ calibration data ("cd") set samples of $n = \{1,…, N\}$ diatom species each, the results of the Q-mode factor analysis are formulated as:

$$U_{cd} = B_{cd}F + E, \qquad (1)$$

where $B_{cd}$ is an $M$ by $L$ varimax rotated factor loadings matrix, $F$ is a factor scores matrix of size $L$ by $N$, $E$ is an error matrix, and $L < N$ denotes the number of factors retained for further analysis. Each element $lm$ of $B_{cd}$, $l = \{1,…, L\}$, represents a proportional contribution of factor $l$ into calibration data set sample $n$. In turn, $F$ describes the species compositions of $L$ varimax rotated factors: each element $F_{ln}$ of $F$ reflects the relative "importance" of species $n$ in factor $l$.

Projecting the row-normalized matrix $U_c$ of downcore diatom assemblages on transposed $F$:

$$B_c = U_cF^t, \qquad (2)$$

yields $K$ by $L$ matrix $B_c$ of the decomposition of $K$ downcore samples into previously defined $L$ varimax factors. Samples communality, found in the main diagonal of matrix $BB^t$ indicates an adequacy of the derived decomposition in terms of the accounted variance.

Andersen et al.[65] provide a detailed description of the eight factors (assemblages) and their spatial distribution. The name of the Sea Ice assemblage in Andersen et al.[65] was re-named to MIZ assemblage in this study as the dominant indicator species in the assemblage, *Fragilariopsis oceanica* (Cleve) Hasle and *Fragilariopsis cylindrus* (Grunow) Krieger in Helmcke and Krieger, dwell in the cold, fresh surface water layer that is produced by melting glacier ice/sea ice. This name reflects more adequately the true (still sea ice related) ecology of these species. These species inhabit both sea ice and open water, but typically form blooms in the melt layer overlying ambient sea water[67],[68]. Hence the MIZ assemblage can be used as an indicator of past melt periods.

**Data availability**. All data presented in this paper are available in the NOAA paleoclimate database and in the open database for Earth and Environmental Science PANGAEA (http://www.ncdc.noaa.gov/paleo; https://www.pangaea.de).

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

## Acknowledgements

We would like thank the captain and all members of the cruise MSM09/2. We also thank O. Hyttinen, University of Helsinki, for her help with grain size distribution analysis. Funding from the Finnish Graduate School in Geology is kindly acknowledged as is the funding from the Academy of Finland (A.E.K. Ojala, QUAL-project 259343). We are grateful to F.M. Nick, H. Machguth, T. Luoto, and K. Pauli for insightful discussions.

## Author contributions

M.O., K.W. and A.M. planned the study, M.O. conducted the diatom analysis, S.J. and D.D. carried out the statistical analyses, R.J. constructed the age-depth model and R.T. contributed with expertise in radiocarbon dating and age modeling, N.J.K. contributed with expertise in glaciology, M.K. organized the cruise as chief scientist and provided the sediment material. All authors contributed to data interpretation and writing of the manuscript.

## Additional information

**Competing interests:** The authors declare no competing financial interests.

