## [Peer Review File · Nature Communications]

Reviewers' comments:

Reviewer #1 (Remarks to the Author):

For many years the Quaternary history of Baffin Bay has been the singular province of Canadian researchers with a series of research cruises on the CSS Hudson starting in the 1970s. In the last decade or so Baffin Bay has been "invaded" by a number of European based research cruises. As I was one of the early researchers on the Quaternary history of Baffin Bay this interest is welcome. The concept of the Younger Dryas (YD) as a major "cold event" largely stems from the glacial history of the Scandinavian and British ice sheets and more recently from the data accumulated in a large number of records from the Greenland Ice Sheet. It is noteworthy however that the evidence for YD around the North American and Greenland ice sheets is limited (Miller, 2008) although there is relatively new evidence for such an event on the West Greenland margin (Jennings et al., 2014b). However, on SE Greenland the ice sheet was at the coastline by YD times (Jennings et al., 2002) and the YD is marked by a major meltwater signal (Jennings et al., 2006) and "mild" conditions on land (Bjorck et al., 2002). Thus the question raised in this submission are the ocean conditions in Baffin Bay during the YD. In Figure 1 the authors need to explain that the currents they portray are in fact the modern conditions. During the YD the input of water from the Arctic Ocean was non-existent because the North American ice sheet extended across the Canadian Arctic Channels, thus we do not really know if there was a "Baffin Current" as it exists today. Certainly we might expect water from East Greenland, including North Atlantic Water, to move into Baffin Bay however the circulation is to an extent an "unknown".

I have several issues with the paper that need to be addressed: first is the very restricted discussion on the ocean reservoir correction, which they take as 140 yr based on the work of Lloyd et al. However, it is not at all clear that the "modern" value is relevant to the YD as the literature around Norway and other areas to the east indicate significant regional variability in the ΔR of several 100 to 1000 yr. I understand that this is a difficult issue but I would point them to the work on the SE Greenland margin that indicates the ΔR was ~ 0 at the time of the Vedde ash (Jennings et al., 2002; Jennings et al., 2014a).

The proxy that the paper uses for their argument of YD retreat are diatoms. These live in the uppermost portion of the water column and the retreat of present day glaciers around Greenland is linked to the incursion of Atlantic Intermediate Water at depth below the upper freshwater coastal current. Can the authors argue with any evidence that the warmer surface water is a measure of the ocean climate say between 10-400 m wd?

It has been known or hypothesized for several decades, starting with the work of Hiscott and Aksu (Hiscott et al., 1989) that the retreat of ice, especially from the ice streams in northern Baffin Bay, is associated with warming conditions. From a stratigraphic viewpoint this is clearly evident by the presence of detrital carbonate-rich (DC) lithofacies (Andrews et al., 2014; Andrews et al., 1998; Simon et al., 2014). In my view it is thus critical for the submission to say whether this lithofacies is evident in their core?

One relevant reference that the paper does not include is from the Baffin Island shelf just south of their site (Andrews et al., 1996), which indicates that the onset of the YD was based on an increase in detrital carbonate and a dramatic increase in glacial marine associated benthic foraminifera. This would suggest that water at 155 m wd at least was cold and probably somewhat fresh.

Conclusion The paper presents data from a well-dated marine core in the southern area of Baffin Bay. The evidence from their specific proxy appears to be well-founded but I think that the paper needs to consider some existing evidence in the light of their conclusion. I get the impression that they are not all that familiar with the glacial history of Baffin Island and the Canadian High Arctic and this does need to be factored in to their story.

I recommend publication with attention to the issues that I have raised above.

Andrews, J.T., Gibb, O.T., Jennings, A.E., Simon, Q., 2014. Variations in the provenance of

sediment from ice sheets surrounding Baffin Bay during MIS 2 and 3 and export to the Labrador Shelf Sea: site HU2008029-0008 Davis Strait. *Journal of Quaternary Science* 29, 3-13.

Andrews, J.T., Kirby, M.E., Aksu, A., Barber, D.C., Meese, D., 1998. Late Quaternary Detrital Carbonate (DC-) events in Baffin Bay (67° - 74° N): Do they correlate with and contribute to Heinrich Events in the North Atlantic? *Quaternary Science Reviews* 17, 1125-1137.

Andrews, J.T., Osterman, L.E., Jennings, A.E., Syvitski, J.P.M., Miller, G.H., Weiner, N., 1996. Abrupt changes in marine conditions, Sunneshine Fiord, eastern Baffin Island, N.W.T. (ca. 66° N) during the last deglacial transition: Links to the Younger Dryas cold-event and Heinrich, H-0, in: Andrews, J.T., Austin, W., Bergsten, H., Jennings, H.E. (Eds.), *Late Quaternary Paleoceanography of North Atlantic Margins*. Geological Society of London, London, pp. 11-27.

Bjorck, S., Bennike, O., Rosen, P., Andresen, C.S., Bohncke, S., Kaas, E., Conley, D., 2002. Anomalously mild Younger Dryas summer conditions in southern Greenland. *Geology* 30, 427-430.

Hiscott, R.N., Aksu, A.E., Nielsen, O.B., 1989. Provenance and dispersal patterns, Pliocene-Pleistocene section at Site 645, Baffin Bay, in: Stewart, S.K.e. (Ed.), *Proceedings ODP, Scientific Results*, 105. Ocean Drilling Program, College Station, TX, pp. 31-52.

Jennings, A.E., Gronvold, K., Hilberman, R., Smith, M., Hald, M., 2002. High resolution study of Icelandic tephra in the Kangerlussuaq Trough, southeast Greenland, during the last deglaciation. *Journal of Quaternary Science* 17, 747-757.

Jennings, A.E., Hald, M., Smith, L.M., Andrews, J.T., 2006. Freshwater forcing from the Greenland Ice Sheet during the Younger Dryas: Evidence from Southeastern Greenland shelf cores. *Quaternary Science Reviews* 25, 282-298.

Jennings, A.E., Thordarson, T., Zalzal, K., Stoner, J.F., Hayward, C., Geirsdottir, A., Miller, G.H., 2014a. Holocene tephra from Iceland and Alaska Record in SE Greenland Shelf sediments, in: Austin, W.E.N., Abbott, P.M., Davis, S., M., Pearce, N.J.G., Wastegard, S. (Eds.), *Marine tephrochronology*. Royal Society of London Special publication 398, pp. 157-193.

Jennings, A.E., Walton, M.E., Cofaigh, C.O., Kilfeather, A., Andrews, J.T., Ortiz, J.D., De Vernal, A., Dowdeswell, J.A., 2014b. Paleoenvironments during Younger Dryas-Early Holocene retreat of the Greenland Ice Sheet from outer Disko Trough, central west Greenland. *Journal of Quaternary Science* 29, 27-40.

Miller, G.H., 2008. Greenland's elusive younger dryas. *Quaternary Science Reviews* 27, 2271-2272.

Simon, Q., Hillaire-Marcel, C., St-Onge, G., Andrews, J.T., 2014. North-eastern Laurentide, western Greenland and southern Innuitian ice stream dynamics during the last glacial cycle. *Journal of Quaternary Science* 29, 14-26.

Reviewer #2 (Remarks to the Author):

Review «Younger Dryas ice margin retreat triggered by warming of the ocean surface in central-eastern Baffin Bay» by Mimmi Oksman et al. submitted to *Nature Communications*.

The manuscript deals with reconstruction of sea surface temperature in central Baffin Bay off Disko Bay in West Greenland c. 14.2-10.2 kyr including the Younger Dryas cold event 12.9-11.7 kyr. The studied core SL 170 is from about 1 km water depth on the western slope of Greenland off Disko Bay. The SST is calculated by transfer functions applied to diatom floras supplemented by factor analysis. The record is c. 6.8 m long and has a high number of radiocarbon dates (21 in total, with three discarded because of age reversals). The material for dates were mainly 'mixed benthic foraminifera', in some cases monospecific samples of the planktic foraminifera *N. pachyderma* (4 samples) and in one case unidentified 'fragments'.

The reconstructed SST varied between 3 and 4.5 °C. The authors conclude that the Younger Dryas stadial showed warm surface conditions in Baffin Bay and that the warming were driven by increased inflow of Atlantic water in combination with the insolation maximum. These findings has a wide interest of the paleoclimate community, but there are several weaknesses of this study that

must be addressed, so to be published much more work is needed. Below I outline my major concerns about this manuscript.

The study is a 'single-proxy' study. It is also a study based on only one record and there has been no attempt to correlate to other records from Baffin Bay or the West Greenland margin with a good chronology with independent time markers. There has also not been any serious attempts to verify the age model or any discussion about changes in reservoir ages during the deglaciation. Being a single-proxy study and relying heavily on radiocarbon ages (modelled ages; original calibrations not shown) that are corrected by the modern reservoir age ($\Delta R=140\pm 35$ yr), I am not convinced that the interval claimed to be Younger Dryas in fact belongs to the Younger Dryas interval. It is well known that reservoir ages were not constant over time and that, in particular during the deglaciation the reservoir ages increased. Off coastal western Norway by a ΔR up to 200 years for the Younger Dryas (Bondevik et al., 2006) at a time when the ice sheet had retreated well into fjords. Over the West Greenland margin, the ice sheet was still very large and reaching to about the mid shelf off Disko (references in manus). Reservoir ages is thus probably much higher than off Norway. Austin et al. (1995) by the aid of well-dated terrestrial tephras in marine records from open ocean and shelf areas including southwest Greenland, found reservoir ages ΔR for the surface water values ranging from 300 to almost 700 years for the Younger Dryas Vedde tephra (10.3 14C kyr in land records; c. 12.17 kyr in Greenland ice core annual layer counts). They found a ΔR value of >300 years for the Pre-Boreal Saksunarvatn tephra (Early Holocene 9.1 kyr in land records, 10.1 kyr in ice core years). There are several other studies of reservoir age changes during the deglaciation. For example, increases in ΔR for surface water of 820 ± 430 to 1010 ± 340 years during the Younger Dryas in open ocean North Atlantic records has been recorded (Waelbroeck et al., 2001). Giving the core site of core SL 170 at high latitudes in Baffin Bay surrounded by huge ice sheets, the planktic ΔR values are bound to be much higher than the present day 140 yrs, not to mention the benthic ΔR values.

Given that reservoir ages are unknown, but probably much higher than the 140 ± 35 yrs used, the Younger Dryas sequence are probably to be found much deeper in the core. The lithological log is no support in pinpointing Younger Dryas, other markers or correlations to other well-dated records with a solid identification of the Younger Dryas is necessary. Best would be to search for the Younger Dryas ASH Zone I tephras; they have been found in southwest Greenland and Labrador Sea and could potentially also be found in Baffin Bay. Another option is looking for detrital carbonate from North Baffin that has been recorded to arrive in the earliest Holocene (e.g., Jennings et al., 2014; Pearce et al., 2015). It would also help to increase the number of proxies to make more correlation possibilities to other studies (XRF-scans, IRD counts and provenance, counts and identification of benthic foraminifera, stable isotope records, prolonging the diatom flora record.....).

Since only model ages are shown (and the calibration program is not indicated, only Marine13) and planktic and benthic dates are mixed, I have re-calibrated some of the given 14C ages from Table 1 in the manus, based on Calib702, Marine13 and a ΔR of 140 ± 35 (2 sigma error; mid-point of age range chosen). The authors can argue that they have many dates mostly in chronological order (after modelling) and that duplicates give much the same ages. I found four small age reversals from the list of 14C ages in Table 1 of the manus. If one accepts the younger of these ages as the correct one, the age at 288-289 cm down core ($12,456\pm 276$ yrs) could be a Younger Dryas age (modelled age of the author: 12,600 yrs). Since the date is benthic and with likely increased ΔR values at least of the order of magnitude found by Austin et al. (1995) and Waelbroeck et al. (2001) (see also Butzin et al., 2005) (they are all surface ΔR values), this date could still be of Holocene age. Then we are left with the date of 13,278 yrs (benthic) and 12,930 yrs (planktic) at 399-402 cm down core (modelled age of the authors: 13,213 yrs) that with some certainty can be argued to be of Younger Dryas age. Thus, the YD interval is probably below 4 m down core. The authors refers to the study of Jennings et al. (2014) based on nearby core records,

and who found no evidence of warming during the Younger Dryas (they had independent time markers in both deep-sea and outer shelf records).

Even if we accepted the age model, the temperature range shown by the transfer functions (Fig. 3a) is 3.2-4.4 °C, a difference of mere 1.2 °C. For the coldest events, 3.2 °C is not very cold as it is a positive temperature and an increase of 1.2 °C is not a lot either (see remarks below about present temperatures in the area). In fact, the RMSE of component 2 is 1.2 °C – thus how certain are the calculated temperatures? And would a 1.2 °C temperature increase cause so much more destabilization of ice sheets? It is therefore essential to see temperature calculations from deeper in the core – when is the temperature increasing to be 3.8 °C (the temperature at the oldest level of the reconstruction)? At some time temperature must have been 'glacial' and sea ice cover more extensive. In other words – the entire sequence analyzed in core SL 170 is not cold at all – it is around modern temperatures for the area (see below).

Another very important point is the lack of CTD data from the research area and comparison to modern conditions. Modern temperatures are not mentioned, and the seasonal variability in temperature and the extent of sea ice cover in modern times are also not mentioned. One has no means of evaluating the past calculated temperatures – are they warmer or colder than today? I found some records showing c. 3 °C at 20 m water depth for July-August just south of the Disko area (another indication that indeed most of the studied record probably is Holocene!). And what are the species composition of diatoms today in the area – and how does it compare with the past floras? How would the 'marginal sea-ice zone' factor, 'Arctic water' factor and 'East-west Greenland current' factor for modern flora analysis look compared to the core record? And what about icebergs and iceberg rafting today?

This brings me to another issue: the factors shown are mostly rather flat curves (Fig. 3c,d), with most clear signals of variability from the 'Marginal Ice Zone' factor (Fig. 3b). I am missing figures that show the actual percentage data of the diatom species? One could select the most abundant species, eventually add together species with similar ecology – then the reader have a better chance of evaluating the results.

Other selected issues:

For the choice of ΔR values five references are given. I checked two (Jennings et al., 2014; Perner et al., 2012). Both these studies cite Lloyd et al. (2011), a study where they checked the Calib database for ΔR values. It would be more appropriate to cite the Lloyd et al. study, not the others.

The calibration program is not mentioned, only Marine13 – is it Calib701? -702?

The 'mixed benthic foraminifera' that are dated – what are the species? One must always be careful by dating whole faunas – miliolid species are notorious for giving too old ages (e.g., Magana et al., 2010). A potential error in addition to the unknown (but probably high) ΔR values.

Conclusions

This high-resolution study of diatom floras certainly have potential, but because of the many and serious shortcomings, the manuscript should be rejected at this stage.

To be convincing, the Younger Dryas time interval must be defined and identified with much greater certainty than demonstrated here and be based on independent time markers such as tephra (if possible) and/or detrital carbonate from Baffin, IRD counts a.o. Changes in ΔR must be discussed and considered.

It is a single-proxy study and more proxies should be included (also to get independent information that can be correlated to other studies). This could be IRD counts, various core logging data (XRF, MSCL data, X-rays...). Stable isotopes performed on benthic/planktic foraminifera could also help in the age model. I also suggest prolonging the diatom record to the bottom of the core (or at least as deep in the core as there are diatoms present). Original percentage data of diatom species should also be shown in figures.

Modern conditions and diatom floras must be presented and the core record compared to the modern situation.

Un-modelled calibrated ages should be shown, and the calibration program used mentioned.

Reviewer #3 (Remarks to the Author):

Based on a high-resolution diatom record, Oksman et al. present a quantitative reconstruction of SST from Baffin bay during the Younger Dryas. Their results support some previous hypothesis that relatively warmer waters (associated to higher seasonality) provoked the collapse of Jakobshavn Isbrae. The results of the authors are interpreted in light of previous work (mostly cited in the ms, but more references are available), but no new interpretation is presented. The originality of their work rely on the quantitative reconstruction of SST at high resolution, which is a valuable data-set. However, it is also the major weakness of the ms: the description of the methods is far too succinct when it is essential to present convincing data. For example, regarding the transfer function, more information are needed: what about the coefficient of determination between observed and inferred values? The value of the maximum bias? Is then SST variation from 3 to 4.5°C significant?

The authors refer too much to previous published work about the transfer function and far much more details should be given, at least in a supplementary information file. How much species are common to the modern data-set and sediment core for example? They discuss past melt-water: any fresh-water diatom species? Only the total variance of the eight factor is indicated: what about the contribution of each one? Are they listed by order of importance? The factor analysis is also confusing for a common reader (and Nature articles are intended for a broader audience): "a Q-mode factor analysis applied to the extended modern diatom calibration data set revealed eight factors": the factor loadings in figure 3 are down-core data.

All along the text, the use of the terms "warm" and "North Atlantic" is also confusing: Miettinen et al. (2015)- presenting the modern data-set- qualify the WGC as temperate, but the authors refers to warm waters from the North Atlantic (or Atlantic water inflow, Atlantic sourced waters...): then, why the factor corresponding to the North Atlantic Current is not also presented? More comments are listed below.

Therefore, in its current state, I would not recommend this manuscript for publication.

Additional comments:

Lines 16-17: the YD was not the only interruption

Line 25: on which basis is an "amplified seasonality" interpreted?

Line 40-41: I guess the authors refer to the Arctic and there are more references on high resolution studies as ref. n°29.

Line 64: what do you intend by large? Number and/or spatial distribution of the samples?

Line 79: it is high resolution indeed, but also a lot of noise and you had to smooth your record.

Why not insisting instead on the quality of the sediments that suggests little bioturbation?

Line 95: reference for the description of the present day distribution of the assemblages? Miettinen et al 2015?

Line 149: is there a fresh-water diatom record?

Line 189: there are more references for the mid-YD

Methods section, coring: I expected here the sampling strategy (not lines 231-232)

Lines 235-236: should cite Miettinen et al as it is a phrase copied from their manuscript!

Lines 248-249: axes listed in different order: explain why please. Why changing the denomination of Andersen et al group? Then, why considering them the equivalent.

Line 250: I don't understand at all this sentence

Lines 251-252: full name of the species are needed

Table 1: specify the fragments

Mixed benthic (duplicate): I assumed they are foraminifera

Figure 2: It would be less confusing if all the description of the chronology would be in the method section

Figure 3: no diatom abundances? It is sometimes referred that they are less abundant in ice/dissolved: would it provide more information?

This reference would have finally enriched the discussion:

Knutz, P. C. et al (2011), Multiple-stage deglacial retreat of the southern Greenland Ice Sheet linked with Irminger Current warm water transport, *Paleoceanography*, 26, PA3204, doi:10.1029/2010PA002053.

Their cores are from nearby and even the YD SST are very limited (but compared to regional data), they discuss the inflow of warm Atlantic waters and the reliability of the warm signal. They also present an interesting IRD record in high-resolution and mention studies arguing for an increased inflow of Atlantic waters to the Nordic seas during the YD.

Response to the reviewers' comments and suggestions

We thank reviewers for their positive comments on our manuscript and appreciate the opportunity to respond to the points raised. The reviewers found the study to have “a wide interest of the paleoclimate community” (Rev#2). The “quantitative reconstruction of SST at a high resolution”, was found to be “a valuable data-set” (Rev#3) and “the evidence from their specific proxy” was stated as “well-founded” (Rev#1). All reviewers agreed that the work has potential to be published in *Nature Communications* after sufficient revision. We would like to thank the reviewers for their insightful and comprehensive reviews which have improved our manuscript. We have addressed all their comments; our responses and changes made to the manuscript are listed below (including references to line and figure numbers in the manuscript). The original comments are in black and our responses in blue.

These are the major changes we made to the manuscript, based on reviewer suggestions:

1. We have added more discussion on the justification of the used reservoir correction age (ΔR) and have strengthened the chronology with detrital carbonate data from Jackson et al. (2017).
 - We created multiple age-depth models by varying only the ΔR age based on reviewer suggestions. We concluded that changing the ΔR to younger (0 years) or older (200-400 years) values does not significantly change the timing of the surface water warming (i.e. warmer SST during the YD), and justify why we discarded the application of the very high reservoir correction (1000 years).
 - We have added a discussion about detrital carbonate layers (DC-events) in the sediment core SL 170 (used in this study and in Jackson et al., 2017). The study by Jackson et al. (2017) linked the timing of these events with other known DC-events in Baffin Bay and Labrador Sea (e.g. Andrews et al., 1995; 2014, Pearce et al., 2013, Jennings et al., 2014, Simon et al., 2012;2014). We emphasize that the use of DC-events as independent time markers has notably strengthened the core chronology.
2. We are now more extensively comparing our results with already existing studies.
 - This was an important issue raised by Reviewer#1 that we addressed in several parts of the revised manuscript.
3. We added additional downcore proxies, extended the diatom record and added diatom concentration data.
 - We conducted grain size distribution analysis that allows us to estimate the vicinity of the meltwater suspension plume (high input of fine-grained sediments) and episodes of IRD deposition (coarser grained sediments). As suggested by Reviewer#2, we have analysed diatom assemblages and reconstructed aSSTs for three additional downcore samples representing ages of 15, 16 and 17 kyr BP based on our age model. We also calculated diatom concentrations on a ca. 100 year resolution throughout the record.

References:

Andrews, J. et al. A Heinrich-Like Event, H-0 (Dc-0) - Source(s) for Detrital Carbonate in the North-Atlantic during the Younger Dryas Chronozone. *Paleoceanography* 10, 943-952 (1995).
Andrews, J.T. et al. Variations in the provenance of sediment from ice sheets surrounding Baffin Bay during MIS 2 and 3 and export to the Labrador Shelf Sea: site HU2008029-0008 Davis Strait. *J. Quat. Sci.* 29, 3-13 (2014).

Jackson et al. Asynchronous instability of the North American-Arctic and Greenland ice sheets during the last deglaciation. *Quat. Sci. Rev.* 164, 140-153 (2017).

Jennings, A. E. et al. Paleoenvironments during Younger Dryas-Early Holocene retreat of the Greenland Ice Sheet from outer Disko Trough, central west Greenland. *J. Quat. Sci.* 29, 27-40 (2014).

Pearce, C. et al. Ocean lead at the termination of the Younger Dryas cold spell. *Nat. Commun.* 4, 1664 (2013).

Simon, Q., St-Onge, G. & Hillaire-Marcel, C. Late Quaternary chronostratigraphic framework of deep Baffin Bay glaciomarine sediments from high-resolution paleomagnetic data. *Geochem. Geophys. Geosyst.* 13, Q0A003 (2012).

Simon, Q. et al. North-eastern Laurentide, western Greenland and southern Innuitian ice stream dynamics during the last glacial cycle. *J. Quat. Sci.* 29, 14-26 (2014).

Reviewers' comments:

Reviewer #1 (Remarks to the Author):

For many years the Quaternary history of Baffin Bay has been the singular province of Canadian researchers with a series of research cruises on the CSS Hudson starting in the 1970s. In the last decade or so Baffin Bay has been "invaded" by a number of European based research cruises. As I was one of the early researchers on the Quaternary history of Baffin Bay this interest is welcome. The concept of the Younger Dryas (YD) as a major "cold event" largely stems from the glacial history of the Scandinavian and British ice sheets and more recently from the data accumulated in a large number of records from the Greenland Ice Sheet. It is noteworthy however that the evidence for YD around the North American and Greenland ice sheets is limited (Miller, 2008) although there is relatively new evidence for such an event on the West Greenland margin (Jennings et al., 2014b). However, on SE Greenland the ice sheet was at the coastline by YD times (Jennings et al., 2002) and the YD is marked by a major meltwater signal (Jennings et al., 2006) and "mild" conditions on land (Bjorck et al., 2002).

Thus the question raised in this submission are the ocean conditions in Baffin Bay during the YD. In Figure 1 the authors need to explain that the currents they portray are in fact the modern conditions. During the YD the input of water from the Arctic Ocean was non-existent because the North American ice sheet extended across the Canadian Arctic Channels, thus we do not really know if there was a "Baffin Current" as it exists today. Certainly we might expect water from East Greenland, including North Atlantic Water, to move into Baffin Bay however the circulation is to an extent an "unknown".

Thank you for pointing this out. We have clarified that Figure 1 illustrates modern ocean currents by adding the word 'modern' in the figure caption.

I have several issues with the paper that need to be addressed: first is the very restricted discussion on the ocean reservoir correction, which they take as 140 yr based on the work of Lloyd et al. However, it is not at all clear that the "modern" value is relevant to the YD as the literature around Norway and other areas to the east indicate significant regional variability in the ΔR of several 100 to 1000 yr. I understand that this is a difficult issue but I would point them to the work on the SE Greenland margin that indicates the ΔR was ~ 0 at the time of the Vedde ash (Jennings et al., 2002; Jennings et al., 2014a).

Assessing reservoir correction ages at high latitudes is indeed a challenging task, especially in the deep marine environment. The ΔR ages used in the previous studies in Baffin Bay are very variable and range between 0 and 400 years (Levac et al., 2001, Knudsen et al. 2008, Ledu et al. 2010). We completely agree with Reviewer#1 that it is crucial to determine the ΔR correction as precisely as possible. Hence, we ran several age-depth models with varying only the ΔR age (0 ± 0 years, 200 ± 100 years, 400 ± 100 years,

1000±100 years and a model with different ΔR for benthic foraminifera (400±50 years) and planktonic foraminifera (140±35 years)), and tested the influence of the resulted chronologies on our aSST -record and on the timing of the detrital carbonate events identified by Jackson et al. (2017). Based on this, we concluded that using the ΔR age of 0 years or increasing the age up to 200 or 400 years does not shift the warm aSST period outside of the YD interval and the onset of the aSST increase remains inside the 95% confidence interval (Fig. S1 Supplementary Information). Only by applying the very high ΔR age of 1000 years would we end up with a significant chronological shift in our reconstruction. This, however, is not supported by the timing – based on several studies – of the Baffin Bay detrital carbonate events also found in core SL 170 (Jackson et al. 2017). We have added the full discussion of the ocean reservoir correction in the Supplementary Information and in the manuscript on lines 155-163.

References:

- Jackson et al. Asynchronous instability of the North American-Arctic and Greenland ice sheets during the last deglaciation. *Quat. Sci. Rev.* 164, 140-153 (2017).
- Knudsen et al. Deglacial and Holocene conditions in northernmost Baffin Bay: sediments, foraminifera, diatoms and stable isotopes. *Boreas* 37, 346-376 (2008).
- Ledu, D. et al. Holocene paleoceanography of the northwest passage, Canadian Arctic Archipelago. *Quat. Sci. Rev.* 29, 3468-3488 (2010).
- Levac, E., De Vernal, A. & Blake, W.Jr. Sea-surface conditions in northernmost Baffin Bay during the Holocene: palynological evidence. *J. Quat. Sci.* 16, 353-363 (2001).

The proxy that the paper uses for their argument of YD retreat are diatoms. These live in the uppermost portion of the water column and the retreat of present day glaciers around Greenland is linked to the incursion of Atlantic Intermediate Water at depth below the upper freshwater coastal current. Can the authors argue with any evidence that the warmer surface water is a measure of the ocean climate say between 10-400 m wd?

Diatoms live in the uppermost 50 meter (sometimes down to ca. 200 m in very clear tropical waters (Round et al. 1990)) of the water column and thus reflect the surface water conditions more than ocean conditions at depth. A CTD profile collected from the coring location on the Disko Shelf in September 2008 (Fig. 1) shows that the modern water column has a relatively warm surface layer (ca. 4°C at 4 m wd) (Krahmann 2013). Below this there is a layer of colder water down to ca. 100 meter of wd, below which temperature and salinity increase, yet remaining slightly colder than the surface waters. The modern WGC is a combination of cold, low-saline polar water originating from the East-Greenland Current and temperate, saline Irminger Current water. These currents meet at the tip of southern Greenland where the two currents are poorly mixed, but mix more efficiently as waters move northwards along the West Greenland (Lloyd 2006). The polar water component of the WGC loses its momentum around the latitude of Fyllas Banke (64°N) and does not penetrate northern Baffin Bay allowing Irminger Current water to rise toward the surface (Krawczyk et al., 2016; Boertmann et al. 2013). Surface waters can thus also (at least partially) reflect intermediate waters. Furthermore, as Jakobshavn Isbræ extended to the open outer shelf during the YD (as opposed to the present day end of a fjord setting), it is not unreasonable to assume that the water circulation close to the glacier was different than the estuarine circulation occurring in fjords occupied by a tidewater glacier today, and also warmer surface waters had access to the glacier front.

Importantly, the diatom species specifically indicating warmer Atlantic water masses are carried by the West Greenland Current to the region, so the overall diatom assemblage recorded in the bottom sediments is a mix of a local assemblage and the species transported with ocean currents to the area.

Parts of the reply above has been added to the manuscript describing modern conditions on lines 48-57.

References:

- Boertmann, D. et al. Disko West. A strategic environmental impact assessment of hydrocarbon activities, Aarhus University, DCE – Danish Centre for Environment and Energy, pp. 306, Scientific Report from DCE – Danish Centre for Environment and Energy No. 71 (2013).
- Krahmann, G. Physical oceanography during Maria S. Merian cruise MSM09/2. IFM-GEOMAR Leibniz-Institute of Marine Sciences, Kiel University, doi:10.1594/PANGAEA.819207 (2013).
- Krawczyk, D.W. et al. Quantitative reconstruction of Holocene sea ice and sea surface temperature off West Greenland from the first regional diatom data set. *Paleoceanography* 31, doi:10.1002/2016PA003003 (2016).
- Lloyd, J.M. Late Holocene environmental changes in Disko Bugt, west Greenland: interaction between climate, ocean circulation and Jakobshavn Isbrae. *Boreas* 35, 35-49 (2006).
- Round, F.E. et al. 1990 *The Diatoms. Biology & Morphology of the Genera*. Cambridge: Cambridge University Press (1990).

It has been known or hypothesized for several decades, starting with the work of Hiscott and Aksu (Hiscott et al., 1989) that the retreat of ice, especially from the ice streams in northern Baffin Bay, is associated with warming conditions. From a stratigraphic viewpoint this is clearly evident by the presence of detrital carbonate-rich (DC) lithofacies (Andrews et al., 2014; Andrews et al., 1998; Simon et al., 2014). In my view it is thus critical for the submission to say whether this lithofacies is evident in their core?

The recent paper by Jackson et al. (2017) presents two DC-lithofacies that are clearly evident in the core SL 170. The older DC-layer has a timing of ca. 14.2-13.7 kyr BP and corresponds to BBDC1 and the younger DC-layer has an age of ca. 12.7-11 kyr BP corresponding to BBDC0. The timing of the younger DC layer agrees well with ice calving from Jakobshavn Isbræ and enhancement of sediment delivery. Based on Jackson et al. (2017) BBDC1 has a stronger northern Baffin Bay signature, whereas BBDC0 is more strongly defined by clastic material from West Greenland (via the retreat of Jakobshavn Isbræ) and a diluted carbonate-rich signal.

References:

- Jackson et al. Asynchronous instability of the North American-Arctic and Greenland ice sheets during the last deglaciation. *Quat. Sci. Rev.* 164, 140-153 (2017).

One relevant reference that the paper does not include is from the Baffin Island shelf just south of their site (Andrews et al., 1996), which indicates that the onset of the YD was based on an increase in detrital carbonate and a dramatic increase in glacial marine associated benthic foraminifera. This would suggest that water at 155 m wd at least was cold and probably somewhat fresh.

The reference, among several others, has been added. Although we don't know exactly what the circulation in Baffin Bay was like during the YD, when the connection to the Arctic Ocean was nonexistent, it appears reasonable to assume that the influence of Atlantic water propagating northward along the West Greenland margin could not be detected on the Baffin Island shelf (similar to modern conditions). Instead, the iceberg and meltwater discharge from northern Baffin Bay associated with Baffin Bay carbonate events (in this case BBDC 0) would explain the glacial marine foraminifera fauna.

Conclusion The paper presents data from a well-dated marine core in the southern area of Baffin Bay. The evidence from their specific proxy appears to be well-founded but I think that the paper needs to consider some existing evidence in the light of their conclusion. I get the impression that they are not all that familiar with the glacial history of Baffin Island and the Canadian High Arctic and this does need to be factored in to their story.

I recommend publication with attention to the issues that I have raised above.

Andrews, J.T., Gibb, O.T., Jennings, A.E., Simon, Q., 2014. Variations in the provenance of sediment from ice sheets surrounding Baffin Bay during MIS 2 and 3 and export to the Labrador Shelf Sea: site HU2008029-0008 Davis Strait. *Journal of Quaternary Science* 29, 3-13.

Andrews, J.T., Kirby, M.E., Aksu, A., Barber, D.C., Meese, D., 1998. Late Quaternary Detrital Carbonate (DC-) events in Baffin Bay (67° - 74° N): Do they correlate with and contribute to Heinrich Events in the North Atlantic? *Quaternary Science Reviews* 17, 1125-1137.

Andrews, J.T., Osterman, L.E., Jennings, A.E., Syvitski, J.P.M., Miller, G.H., Weiner, N., 1996. Abrupt changes in marine conditions, Sunneshine Fiord, eastern Baffin Island, N.W.T. (ca. 66° N) during the last deglacial transition: Links to the Younger Dryas cold-event and Heinrich, H-0, in: Andrews, J.T., Austin, W., Bergsten, H., Jennings, H.E. (Eds.), *Late Quaternary Paleoceanography of North Atlantic Margins*. Geological Society of London, London, pp. 11-27.

Bjorck, S., Bennike, O., Rosen, P., Andresen, C.S., Bohncke, S., Kaas, E., Conley, D., 2002. Anomalously mild Younger Dryas summer conditions in southern Greenland. *Geology* 30, 427-430.

Hiscott, R.N., Aksu, A.E., Nielsen, O.B., 1989. Provenance and dispersal patterns, Pliocene-Pleistocene section at Site 645, Baffin Bay, in: Stewart, S.K.e. (Ed.), *Proceedings ODP, Scientific Results*, 105. Ocean Drilling Program, College Station, TX, pp. 31-52.

Jennings, A.E., Gronvold, K., Hilberman, R., Smith, M., Hald, M., 2002. High resolution study of Icelandic tephra in the Kangerlussuaq Trough, southeast Greenland, during the last deglaciation. *Journal of Quaternary Science* 17, 747-757.

Jennings, A.E., Hald, M., Smith, L.M., Andrews, J.T., 2006. Freshwater forcing from the Greenland Ice Sheet during the Younger Dryas: Evidence from Southeastern Greenland shelf cores. *Quaternary Science Reviews* 25, 282-298.

Jennings, A.E., Thordarson, T., Zalzal, K., Stoner, J.F., Hayward, C., Geirsdottir, A., Miller, G.H., 2014a. Holocene tephra from Iceland and Alaska Record in SE Greenland Shelf sediments, in: Austin, W.E.N., Abbott, P.M., Davis, S., M., Pearce, N.J.G., Wastegard, S. (Eds.), *Marine tephrochronology*. Royal Society of London Special publication 398, pp. 157-193.

Jennings, A.E., Walton, M.E., Cofaigh, C.O., Kilfeather, A., Andrews, J.T., Ortiz, J.D., De Vernal, A., Dowdeswell, J.A., 2014b. Paleoenvironments during Younger Dryas-Early Holocene retreat of the Greenland Ice Sheet from outer Disko Trough, central west Greenland. *Journal of Quaternary Science* 29, 27-40.

Miller, G.H., 2008. Greenland's elusive younger dryas. *Quaternary Science Reviews* 27, 2271-2272.

Simon, Q., Hillaire-Marcel, C., St-Onge, G., Andrews, J.T., 2014. North-eastern Laurentide, western Greenland and southern Inuitian ice stream dynamics during the last glacial cycle. *Journal of Quaternary Science* 29, 14-26.

Reviewer #2 (Remarks to the Author):

Review «Younger Dryas ice margin retreat triggered by warming of the ocean surface in central-eastern Baffin Bay» by Mimmi Oksman et al. submitted to Nature Communications.

The manuscript deals with reconstruction of sea surface temperature in central Baffin Bay off Disko Bay in West Greenland c. 14.2-10.2 kyr including the Younger Dryas cold event 12.9-11.7 kyr. The studied core SL 170 is from about 1 km water depth on the western slope of Greenland off Disko Bay. The SST is calculated by transfer functions applied to diatom floras supplemented by factor analysis. The record is c. 6.8 m long

and has a high number of radiocarbon dates (21 in total, with three discarded because of age reversals). The material for dates were mainly 'mixed benthic foraminifera', in some cases monospecific samples of the planktic foraminifera *N. pachyderma* (4 samples) and in one case unidentified 'fragments'. The reconstructed SST varied between 3 and 4.5 °C. The authors conclude that the Younger Dryas stadial showed warm surface conditions in Baffin Bay and that the warming were driven by increased inflow of Atlantic water in combination with the insolation maximum. These findings has a wide interest of the paleoclimate community, but there are several weaknesses of this study that must be addressed, so to be published much more work is needed. Below I outline my major concerns about this manuscript.

The study is a 'single-proxy' study. It is also a study based on only one record and there has been no attempt to correlate to other records from Baffin Bay or the West Greenland margin with a good chronology with independent time markers. There has also not been any serious attempts to verify the age model or any discussion about changes in reservoir ages during the deglaciation.

Being a single-proxy study and relying heavily on radiocarbon ages (modelled ages; original calibrations not shown) that are corrected by the modern reservoir age ($\Delta R=140\pm 35$ yr), I am not convinced that the interval claimed to be Younger Dryas in fact belongs to the Younger Dryas interval. It is well known that reservoir ages were not constant over time and that, in particular during the deglaciation the reservoir ages increased. Off coastal western Norway by a ΔR up to 200 years for the Younger Dryas (Bondevik et al., 2006) at a time when the ice sheet had retreated well into fjords. Over the West Greenland margin, the ice sheet was still very large and reaching to about the mid shelf off Disko (references in manus). Reservoir ages is thus probably much higher than off Norway. Austin et al. (1995) by the aid of well-dated terrestrial tephras in marine records from open ocean and shelf areas including southwest Greenland, found reservoir ages ΔR for the surface water values ranging from 300 to almost 700 years for the Younger Dryas Vedde tephra (10.3 14C kyr in land records; c. 12.17 kyr in Greenland ice core annual layer counts). They found a ΔR value of >300 years for the Pre-Boreal Saksunarvatn tephra (Early Holocene 9.1 kyr in land records, 10.1 kyr in ice core years). There are several other studies of reservoir age changes during the deglaciation. For example, increases in ΔR for surface water of 820 ± 430 to 1010 ± 340 years during the Younger Dryas in open ocean North Atlantic records has been recorded (Waelbroeck et al., 2001). Giving the core site of core SL 170 at high latitudes in Baffin Bay surrounded by huge ice sheets, the planktic ΔR values are bound to be much higher than the present day 140 yrs, not to mention the benthic ΔR values.

This is indeed a very valid concern and we have significantly improved our discussion on the ocean reservoir correction which is added in Supplementary Information and on lines 155-163 in the manuscript. We are aware that the reservoir age has varied through time and thus ran several age-depth models varying only the ΔR age in order to see the influence of the resulted chronologies on the timing of events in our aSST-record and on the detrital carbonate events (Jackson et al. 2017). We ran five additional age-depth models varying the ΔR age between 0 and 400 years based on previous chronologies from Baffin Bay (e.g. Levac et al., 2001, Knudsen et al. 2008, Ledu et al. 2010) and tested the influence of a very high ΔR (1000 ± 100 years). We found that applying a higher reservoir correction does not shift the warm aSST period outside of the YD interval and the onset of the aSST increase remains inside the 95% confidence interval (Fig. S1). Only by increasing the ΔR age to the very high 1000 ± 100 years, the timing of the surface water warming post-dates the YD interval. This very high ΔR age also shifts the age of observed DC-layers and results in an early Holocene age for the latter DC-layer, occurring after the main discharge events in the Disko Bay region (e.g. Lloyd et al. 2005). It should be further noted that no Holocene-age (<10 000 years) DC-events have been reported from Baffin Bay (Simon et al. 2014; 2012, Jennings et al. 2014).

References:

- Jackson et al. Asynchronous instability of the North American-Arctic and Greenland ice sheets during the last deglaciation. *Quat. Sci. Rev.* 164, 140-153 (2017).
- Jennings, A. E. et al. Paleoenvironments during Younger Dryas-Early Holocene retreat of the Greenland Ice Sheet from outer Disko Trough, central west Greenland. *J. Quat. Sci.* 29, 27-40 (2014).
- Knudsen et al. Deglacial and Holocene conditions in northernmost Baffin Bay: sediments, foraminifera, diatoms and stable isotopes. *Boreas* 37, 346-376 (2008).
- Ledu, D. et al. Holocene paleoceanography of the northwest passage, Canadian Arctic Archipelago. *Quat. Sci. Rev.* 29, 3468-3488 (2010).
- Levac, E., De Vernal, A. & Blake, W.Jr. Sea-surface conditions in northernmost Baffin Bay during the Holocene: palynological evidence. *J. Quat. Sci.* 16, 353-363 (2001).
- Lloyd, J. et al. Early holocene palaeoceanography and deglacial chronology of Disko Bugt, West Greenland. *Quat. Sci. Rev.* 24, 1741-1755 (2005).
- Simon, Q. et al. North-eastern Laurentide, western Greenland and southern Innuitian ice stream dynamics during the last glacial cycle. *J. Quat. Sci.* 29, 14-26 (2014).
- Simon, Q. et al. Late Quaternary chronostratigraphic framework of deep Baffin Bay glaciomarine sediments from high-resolution paleomagnetic data. *Geochem. Geophys. Geosyst.* 13, Q0A003 (2012).

Given that reservoir ages are unknown, but probably much higher than the 140 ± 35 yrs used, the Younger Dryas sequence are probably to be found much deeper in the core. The lithological log is no support in pinpointing Younger Dryas, other markers or correlations to other well-dated records with a solid identification of the Younger Dryas is necessary. Best would be to search for the Younger Dryas ASH Zone I tephtras; they have been found in southwest Greenland and Labrador Sea and could potentially also be found in Baffin Bay. Another option is looking for detrital carbonate from North Baffin that has been recorded to arrive in the earliest Holocene (e.g., Jennings et al., 2014; Pearce et al., 2015).

We did not find tephtras or pollen (i.e. date markers independent of reservoir correction) in our core. We would also like to point out that tephtra layers found in marine sediment cores around Greenland are often used as proxy for ice rafting since they are stored in the ice for a long time and do generally not represent primary atmospheric deposition of tephtras (Jennings et al., 2002, Knutz et al., 2011, Jennings et al., 2014).

Our chronology is significantly strengthened by detrital carbonate events identified in the core SL 170 (Jackson et al., 2017) that were found to correspond with recorded Baffin Bay Detrital Carbonate Events (BBDC-events). The older detrital carbonate event has a timing of ca. 14.2-13.7 kyr BP corresponding to BBDC1 (Simon et al., 2012; 2014, Andrews et al., 2014) and the younger has a timing of ca. 12.7-11 kyr BP corresponding to BBDC0 (Andrews et al., 1995; 1996 Simon et al., 2012;2014, Jennings et al., 2014).

References

- Andrews, J. et al. A Heinrich-Like Event, H-0 (Dc-0) - Source(s) for Detrital Carbonate in the North-Atlantic during the Younger Dryas Chronozone. *Paleoceanography* 10, 943-952 (1995).
- Andrews, J. T. et al. Abrupt changes in marine conditions, Sunneshine Fjord, eastern Baffin Island, NWT during the last deglacial transition: Younger Dryas and H-0 events in: Andrews, J.T. et al. (Eds.), *Late Quaternary Paleocanography of North Atlantic Margins*. Geological Society Special Publication No. 111, 11-27 (1996).
- Andrews, J.T. et al. Variations in the provenance of sediment from ice sheets surrounding Baffin Bay during MIS 2 and 3 and export to the Labrador Shelf Sea: site HU2008029-0008 Davis Strait. *J. Quat. Sci.* 29, 3-13 (2014).
- Jennings, A. et al. High-resolution study of Icelandic tephtras in the Kangerlussuaq Trough, southeast Greenland, during the last deglaciation. *J. Quat. Sci.* 17, 747-757 (2002).
- Jennings, A. E. et al. Paleoenvironments during Younger Dryas-Early Holocene retreat of the Greenland Ice Sheet from outer Disko Trough, central west Greenland. *J. Quat. Sci.* 29, 27-40 (2014).

Knutz, P. C. et al. Multiple-stage deglacial retreat of the southern Greenland Ice Sheet linked with Irminger Current warm water transport. *Paleoceanography* 26, PA3204 (2011).

Simon, Q., St-Onge, G. & Hillaire-Marcel, C. Late Quaternary chronostratigraphic framework of deep Baffin Bay glaciomarine sediments from high-resolution paleomagnetic data. *Geochem. Geophys. Geosyst.* 13, Q0A003 (2012).

Simon, Q. et al. North-eastern Laurentide, western Greenland and southern Innuitian ice stream dynamics during the last glacial cycle. *J. Quat. Sci.* 29, 14-26 (2014).

It would also help to increase the number of proxies to make more correlation possibilities to other studies (XRF-scans, IRD counts and provenance, counts and identification of benthic foraminifera, stable isotope records, prolonging the diatom flora record.....).

As suggested, we have improved our manuscript by analyzing additional proxies. We measured grain size distribution and have also calculated diatom concentrations at a ca. 100 year-resolution (Fig. S2). We also extended the diatom record and analyzed three samples from 15, 16 and 17 kyr BP pre-dating the time period included in this study (Fig. S2). Finally, we have added a reference to a recently accepted paper (Jackson et al. 2017) that presents XRF-scans and discusses them in details together with other physical sediment properties of core SL 170.

References:

Jackson et al. Asynchronous instability of the North American-Arctic and Greenland ice sheets during the last deglaciation. *Quat. Sci. Rev.* 164, 140-153 (2017).

Since only model ages are shown (and the calibration program is not indicated, only Marine13) and planktic and benthic dates are mixed, I have re-calibrated some of the given ^{14}C ages from Table 1 in the manus, based on Calib702, Marine13 and a ΔR of 140 ± 35 (2 sigma error; mid-point of age range chosen). The authors can argue that they have many dates mostly in chronological order (after modelling) and that duplicates give much the same ages. I found four small age reversals from the list of ^{14}C ages in Table 1 of the manus. If one accepts the younger of these ages as the correct one, the age at 288-289 cm down core ($12,456 \pm 276$ yrs) could be a Younger Dryas age (modelled age of the author: 12,600 yrs). Since the date is benthic and with likely increased ΔR values at least of the order of magnitude found by Austin et al. (1995) and Waelbroeck et al. (2001) (see also Butzin et al., 2005) (they are all surface ΔR values), this date could still be of Holocene age. Then we are left with the date of 13,278 yrs (benthic) and 12,930 yrs (planktic) at 399-402 cm down core (modelled age of the authors: 13,213 yrs) that with some certainty can be argued to be of Younger Dryas age. Thus, the YD interval is probably below 4 m down core. The authors refers to the study of Jennings et al. (2014) based on nearby core records, and who found no evidence of warming during the Younger Dryas (they had independent time markers in both deep-sea and outer shelf records).

The radiocarbon ages were calibrated using the CALIB Rev 7.0.4 program (Stuiver and Reimer, 1993) using Marine13 calibration curve (Reimer et al., 2013) and reservoir correction (ΔR) of 140 ± 35 years. This is now mentioned in the manuscript on lines 273-274 and Table 1. For the concerns regarding the ΔR values, please see our earlier replies.

References:

Reimer, P. J. et al., IntCal13 and Marine13 radiocarbon age calibration curves 0-50,000 years cal BP. *Radiocarbon* 55, 1869-1887, doi:10.2458/azu_js_rc.55.16947 (2013).

Stuiver, M. and Reimer, P.J. Extended ^{14}C database and revised CALIB radiocarbon calibration program. *Radiocarbon* 35, 215-230 (1993).

Even if we accepted the age model, the temperature range shown by the transfer functions (Fig. 3a) is 3.2-4.4 °C, a difference of mere 1.2 °C. For the coldest events, 3.2 °C is not very cold as it is a positive

temperature and an increase of 1.2 °C is not a lot either (see remarks below about present temperatures in the area). In fact, the RMSE of component 2 is 1.2 °C – thus how certain are the calculated temperatures?

The performance of the WA-PLS transfer function has been estimated using a *h*-block cross-validation (Trachsel, M. & Telford, R. J. 2016). This results in slightly higher error estimates for the WA-PLS model (RMSE 1.14°C, r^2 0.92) compared to other cross-validation methods. The higher prediction error is due to the fact that this cross-validation method accounts for spatial dependency, whereas some studies use the leave-one-out cross-validation (jackknife) (e.g. Miettinen et al., 2011; 2012, Krawczyk et al. 2016) that does not allow for spatial autocorrelation and underestimates the prediction error because of effective pseudoreplication. Strictly, the RMSE for the transfer function is an estimate of the uncertainty in the absolute value of (statistically) independent fossil samples and depends, in part, on the modern calibration (training) set used. The core samples are a temporal sequence and not statistically independent, so while the overall absolute error of the sequence is of a similar magnitude to the reconstructed temperature change, the reconstructed trends and amplitudes themselves are real and robust.

References:

Krawczyk, D.W. et al. Quantitative reconstruction of Holocene sea ice and sea surface temperature off West Greenland from the first regional diatom data set. *Paleoceanography* 31, doi:10.1002/2016PA003003 (2016).
Miettinen, A. et al. North Atlantic sea surface temperatures and their relation to the North Atlantic Oscillation during the last 230 years. *Clim. Dyn.* 36, 533-543 (2011).
Miettinen A. et al. Multicentennial variability of the sea surface temperature gradient across the subpolar North Atlantic over the last 2.8 kyr. *Journal of Climate* 25, 4205–4219 (2012).
Trachsel, M. & Telford, R. J. Technical note: Estimating unbiased transfer-function performances in spatially structured environments. *Clim. Past* 12, 1215-1223 (2016).

And would a 1.2 °C temperature increase cause so much more destabilization of ice sheets?

A SST temperature difference of ~1.2 °C has been associated to regional ocean terminating glacier retreat rates similar to the retreat rates of Jakobshavn Isbrae after the YD readvance (Kjær et al., 2012; Bjørk et al., 2012). Notably, Bjørk et al. (2012) show that the marine terminating glaciers in southeast Greenland responded strongly in the 2000s to change in SST (anomalies), which Murray et al. (2010, fig. 5) suggest is the first-order regional control on this retreat. The arrival of a warm water pulse on the shelf in Disko Bay in 1997 increased the mean temperatures from 1.7 to 3.3 °C (differences of 1.1 to 1.6 °C depending on depth), and caused Jakobshavn Isbrae to accelerate, thin, and retreat (Holland et al., 2008; more in Motyka et al, 2011). Dynamic thinning of Jakobshavn Isbrae has now propagated more than a hundred of kilometers into the interior of the GrIS (e.g. Kjeldsen et al., 2015, fig. 2i).

A mechanism for the YD advance is proposed by Ó Cofaigh et al., (2013) where Jakobshavn ice stream surges, thereby thinning the glacier and causing a lowering of the surface profile via dynamic thinning. Indeed, a thin outlet with a low surface profile is susceptible to retreat since the lower surface profile means that much less ice is moved forward by the ice stream and a rapid retreat mainly by calving will follow.

References:

Bjørk, A.A. et al. An aerial view of 80 years of climate-related glacier fluctuations in southeast Greenland. *Nat. Geosci.* 5, 427-432 (2012).
Holland, D.M. et al. Acceleration of Jakobshavn Isbrae triggered by warm subsurface ocean waters. *Nat. Geosci.* 1, 659-664 (2008).
Kjeldsen, K.K. et al. Spatial and temporal distribution of mass loss from the Greenland Ice Sheet since AD 1900. *Nature* 528, 396–400 (2015).

Kjær, K.H. et al. Aerial Photographs Reveal Late-20th-Century Dynamic Ice Loss in Northwestern Greenland. *Science* 337, 569-573 (2012).

Motyka R.J. et al. Submarine melting of the 1985 Jakobshavn Isbræ floating tongue and the triggering of the current retreat. *J. Geophys. Res. Earth Surf.* 116, 1-17 (2011).

Murray, T. et al. Ocean regulation hypothesis for glacier dynamics in southeast Greenland and implications for ice sheet mass changes. *J. Geophys. Res.*, 115 1-15 (2010).

Ó Cofaigh, C. et al. An extensive and dynamic ice sheet on the West Greenland shelf during the last glacial cycle. *Geology* 41, 219-222 (2013).

It is therefore essential to see temperature calculations from deeper in the core – when is the temperature increasing to be 3.8 °C (the temperature at the oldest level of the reconstruction)? At some time temperature must have been ‘glacial’ and sea ice cover more extensive. In other words – the entire sequence analyzed in core SL 170 is not cold at all – it is around modern temperatures for the area (see below).

As already mentioned above, we extended the diatom record by analyzing additional three samples (15, 16 and 17 kyr BP, which is at the bottom of the dated interval (Fig 2, no datable material was found deeper in the core). The aSSTs reconstructed for these samples are 3.4°C, 3.6°C and 3.5°C suggesting an increasing trend towards 3.8°C (Fig. S2).

However, the temperatures reconstructed here are well below the modern average summer (July) surface temperature of 5°C (Boertmann et al., 2013).

Again, we would like to stress that the reconstructed values for any environmental variable - when using transfer functions - depends on the range of the environmental variable in the used modern calibration set. While WA-PLS is able to reconstruct beyond these calibration data set boundaries (here the SST range of the calibration data set is 0.0-19.7°C), it is still to some extent affected by the issue. Our training set has relatively few samples (8) with aSST <2°C so our reconstructed temperatures at low aSST may be slightly higher than the real values would have been. However, as mentioned earlier, the trends and their amplitude are reliable.

References:

Boertmann, D. et al. Disko West. A strategic environmental impact assessment of hydrocarbon activities, Aarhus University, DCE – Danish Centre for Environment and Energy, pp. 306, Scientific Report from DCE – Danish Centre for Environment and Energy No. 71 (2013).

Another very important point is the lack of CTD data from the research area and comparison to modern conditions. Modern temperatures are not mentioned, and the seasonal variability in temperature and the extent of sea ice cover in modern times are also not mentioned. One has no means of evaluating the past calculated temperatures – are they warmer or colder than today? I found some records showing c. 3 °C at 20 m water depth for July-August just south of the Disko area (another indication that indeed most of the studied record probably is Holocene!).

We fully agree that it is important to also mention the modern conditions and have added a description of the modern conditions (temperature and sea ice cover) as well as a CTD profile from the coring location into the manuscript (Fig. 1, lines 41-44 and 48-57). As mentioned above, the modern summer (July) water temperatures in the area are on average around 5°C (Boertmann et al., 2013), which is higher than the temperatures we have reconstructed.

References:

Boertmann, D. et al. Disko West. A strategic environmental impact assessment of hydrocarbon activities, Aarhus University, DCE – Danish Centre for Environment and Energy, pp. 306, Scientific Report from DCE – Danish Centre for Environment and Energy No. 71 (2013).

Krahmann, G. Physical oceanography during Maria S. Merian cruise MSM09/2. IFM-GEOMAR Leibniz-Institute of Marine Sciences, Kiel University, doi:10.1594/PANGAEA.819207 (2013).

And what are the species composition of diatoms today in the area – and how does it compare with the past floras? How would the ‘marginal sea-ice zone’ factor, ‘Arctic water’ factor and ‘East-west Greenland current’ factor for modern flora analysis look compared to the core record?

The modern Baffin Bay diatom flora is described in e.g. Williams (1986), Sha et al. (2014), Krawczyk et al (2016), whereas descriptions of Holocene and older (back to ca. 11 000 years ago) diatom flora from the area can be found in e.g. Moros et al. (2006), Ren et al. (2009), Sha et al. (2014), Krawczyk et al. (2013; 2016). The assemblages in these studies are very similar to the assemblages in core SL 170.

We have further compared modern diatom assemblages with the down-core assemblages in the Supplementary Information. Figures S2 and S5 present respectively the 14 most abundant species found in the core SL 170 and the relative abundance of the same species in 12 modern surface samples from Baffin Bay (included in the used diatom data set (Miettinen et al. 2015)). Both records show dominance of the same species, although there are differences in the relative abundances of individual species (see Supplementary Information). Hence, based on already published work and our current study, the species composition has been the same from deglacial times to present, only the relative abundances of the species in the assemblages have changed depending on the prevailing environmental conditions.

We ran the factor analysis for 12 surface samples in the dataset that are taken from the Baffin Bay and the Labrador Sea area (65-75°N) (Table S2). The analysis shows that the Baffin Bay and the Labrador Sea are associated with a regional dominance of the Arctic water, the Marginal Ice Zone assemblages and with a minor contribution of the Greenland Arctic Water and the East-West Greenland Current assemblages with a pronounced southward shift in the maximum loading from the Marginal Ice Zone and East-West Greenland Currents factors.

We also like to stress that the factors are calculated based on modern ocean conditions, relating diatom assemblages from modern surface sediment samples to modern surface hydrography (Andersen et al., 2004). Past surface hydrography/factors are then inferred from diatom species found in the core. The high communality (i.e. the proportion of each species’ variance that can be explained by the factors), for the down core samples support our approach.

References:

Andersen, C. et al. Nonuniform response of the major surface currents in the Nordic Seas to insolation forcing: Implications for the Holocene climate variability. *Paleoceanography* 19, PA2003 (2004).

Krawczyk, D.W. et al. Late-Holocene diatom derived seasonal variability in hydrological conditions off Disko Bay, West Greenland. *Quat. Sci. Rev.* 67, 93-104 (2013).

Krawczyk, D.W. et al. Quantitative reconstruction of Holocene sea ice and sea surface temperature off West Greenland from the first regional diatom data set. *Paleoceanography* 31, doi:10.1002/2016PA003003 (2016).

Moros, M. et al. Mid- to late-Holocene hydrological and climatic variability in Disko Bugt, central West Greenland. *The Holocene* 16, 357-367 (2006).

Ren, J. et al. A diatom-based reconstruction of Early Holocene hydrographic and climatic change in a southwest Greenland fjord. *Mar. Micropal.* 70, 166-176 (2009).

Sha, L. et al. A diatom-based sea-ice reconstruction for the Vaigat Strait (Disko Bugt, West Greenland) over the last 5000 yr. *Palaeogeography, Palaeoclimatology, Palaeoecology* 403, 66–79 (2014).

Williams, K.M. Recent Arctic marine diatom assemblages from bottom sediments in Baffin Bay and Davis Strait. *Mar. Micropaleontol.* 10, 327-341 (1986).

And what about icebergs and iceberg rafting today?

Weidick and Bennike (2007) and Joughin et al. (2004) have shown that Jakobshavn Isbræ has a significant role in Greenland iceberg production, since it produces yearly ca. 35-50 km³ of icebergs that is over 10% of the total iceberg output from the Greenland Ice Sheet and thus Jakobshavn Isbræ's calf-ice production has particular importance for the mass balance of the Greenland ice sheet. A study by Andresen et al. (2012) shows that most of the calving occurs under a strong Atlantic water influence and under the warm oceanic conditions most of IRD is released close to the Fjord. Yet, many of the Disko Bay icebergs will travel to the Labrador Sea through Davis Strait, whereas icebergs produced from glaciers north of Disko Bay may be carried northward along the West Greenland Current (Tang 2004).

We have mentioned in the manuscript that Jakobshavn Isbræ produces 10% of the total iceberg discharge from the Greenland Ice Sheet (line 66).

References:

Andresen, C. S. et al. Rapid response of Helheim Glacier in Greenland to climate variability over the past century. *Nat. Geosci.* 5, 37-41 (2012).

Joughin, I., Abdalati, W. and Fahnestock, M. Large fluctuations in speed on Greenland's Jakobshavn Isbræ glacier. *Nature* 432, 608–610 (2004).

Tang, C.C.L. et al. The circulation, water masses and sea-ice of Baffin Bay. *Progress in Oceanography* 63, 183-228 (2004).

Weidick, A. and Bennike, O. Quaternary glaciation history and glaciology of Jakobshavn Isbræ and the Disko Bugt region, West Greenland: a review. *Geol. Surv. Den. Green. Bull.* 14, 78 pp.

This brings me to another issue: the factors shown are mostly rather flat curves (Fig. 3c,d), with most clear signals of variability from the 'Marginal Ice Zone' factor (Fig. 3b). I am missing figures that show the actual percentage data of the diatom species? One could select the most abundant species, eventually add together species with similar ecology – then the reader have a better chance of evaluating the results.

We have added figure S2 in the Supplementary Information showing the diatom percentage data. We would, however, stress that factor analysis is a powerful tool as it quantitatively relates diatom species distribution to modern surface ocean hydrography over large spatial scales (in this case the whole North Atlantic).

Other selected issues:

For the choice of ΔR values five references are given. I checked two (Jennings et al., 2014; Perner et al., 2012). Both these studies cite Lloyd et al. (2011), a study where they checked the Calib database for ΔR values. It would be more appropriate to cite the Lloyd et al. study, not the others.

We have changed the citation only to Lloyd et al. (2011) and have mentioned other studies using the same ΔR age based on Lloyd et al. 2011.

The calibration program is not mentioned, only Marine13 – is it Calib701? -702?

As mentioned earlier, the radiocarbon ages were calibrated using the CALIB Rev 7.0.4 program (Stuiver and Reimer, 1993) which we have added to the manuscript on lines 273-274.

References:

Stuiver, M. and Reimer, P.J. Extended 14C database and revised CALIB radiocarbon calibration program. *Radiocarbon* 35, 215-230 (1993).

The 'mixed benthic foraminifera' that are dated – what are the species? One must always be careful by dating whole faunas – miliolid species are notorious for giving too old ages (e.g., Magana et al., 2010). A potential error in addition to the unknown (but probably high) ΔR values.

The benthic foraminifera assemblages used for radiocarbon dating included the species *Cassadulina reniforme*, *Cassadulina neoteretis*, *Elphidium excavatum*, *Melonis barleeanus*, *Astrononion gallowayi* and *Islandiella norcrossi*. Due to the varying contributions of these species to the assemblage downcore, mixed samples were required to provide enough carbonate for radiocarbon dating. Benthic assemblage analysis indicate that miliolid spp. were scarce down core (Jackson, R. personal communication). On the rare occasion when miliolid spp. were found they were not included in the samples sent for radiocarbon dating.

We have added the text above to the Methods section of the manuscript.

Conclusions

This high-resolution study of diatom floras certainly have potential, but because of the many and serious shortcomings, the manuscript should be rejected at this stage.

To be convincing, the Younger Dryas time interval must be defined and identified with much greater certainty than demonstrated here and be based on independent time markers such as tephra (if possible) and/or detrital carbonate from Baffin, IRD counts a.o. Changes in ΔR must be discussed and considered. It is a single-proxy study and more proxies should be included (also to get independent information that can be correlated to other studies). This could be IRD counts, various core logging data (XRF, MSCL data, X-rays....). Stable isotopes performed on benthic/planktic foraminifera could also help in the age model. I also suggest prolonging the diatom record to the bottom of the core (or at least as deep in the core as there are diatoms present). Original percentage data of diatom species should also be shown in figures. Modern conditions and diatom floras must be presented and the core record compared to the modern situation.

Un-modelled calibrated ages should be shown, and the calibration program used mentioned.

Please find detailed comments to the issue summarized here at relevant places earlier in the response letter.

Reviewer #3 (Remarks to the Author):

Based on a high-resolution diatom record, Oksman et al. present a quantitative reconstruction of SST from Baffin bay during the Younger Dryas. Their results support some previous hypothesis that relatively warmer waters (associated to higher seasonality) provoked the collapse of Jakobshavn Isbrae. The results of the authors are interpreted in light of previous work (mostly cited in the ms, but more references are available), but no new interpretation is presented.

The originality of their work rely on the quantitative reconstruction of SST at high resolution, which is a valuable data-set. However, it is also the major weakness of the ms: the description of the methods is far too succinct when it is essential to present convincing data. For example, regarding the transfer function,

more information are needed: what about the coefficient of determination between observed and inferred values? The value of the maximum bias? Is then SST variation from 3 to 4.5°C significant?

We have now added a more detailed description of the method in the manuscript (lines 288-293) and in the Supplementary Information. The performance of the WA-PLS model has been estimated with *h*-block cross-validation resulting in an RMSE of 1.14°C, a coefficient of determination between observed and inferred values (r^2) of 0.92 and a maximum bias of 2.8°C. Reviewer#1 also questions the significance of the of the SST variation from 3 to 4.5 °C. Our reply is the same as above - the RMSE for the transfer function is an estimate of the error when the model is applied to new and (statistically) independent fossil samples. The core samples are a temporal sequence and are not statistically independent so while the overall absolute error of the sequence is of a similar magnitude to the reconstructed temperature change, the reconstructed trends and amplitudes are themselves real and robust and reflect trends which are clearly visible in the underlying diatom biostratigraphy.

The authors refer too much to previous published work about the transfer function and far much more details should be given, at least in a supplementary information file. How much species are common to the modern data-set and sediment core for example?

As mentioned above, we have added a more detailed description of the transfer function in the manuscript (lines 288-293) and in the Supplementary Information. We have also added a quantitative comparison between the core and calibration set diatom species in the Supplementary Information. In addition, we want to emphasize that the communality metric in the factor analysis (Fig. S4) shows that ca. 80% of the variance on average in the downcore assemblages can be "explained" using the spatial variability in the modern calibration dataset.

They discuss past melt-water: any fresh-water diatom species?

Despite large melt events, surface salinities would still be relatively high (generally close to or above 30 per mil, see e.g. Straneo et al. 2012 for current glacier-proximal locations). Hence one wouldn't find any fresh water diatoms due to melt events. In addition, true fresh water diatoms are usually only found in coastal settings, where rivers and streams enter the sea.

References:

Straneo, F., Sutherland, D. A., Holland, D., Gladish, C., Hamilton, G. S., Johnson, H. L., Rignot, E., Xu, Y. and Koppes, M. Characteristics of ocean waters reaching Greenland's glaciers. *Annals of Glaciology* 53(60), 202-210, doi:10.3189/2012AoG60A059 (2012).

Only the total variance of the eight factor is indicated: what about the contribution of each one? Are they listed by order of importance? The factor analysis is also confusing for a common reader (and Nature articles are intended for a broader audience): "a Q-mode factor analysis applied to the extended modern diatom calibration data set revealed eight factors": the factor loadings in figure 3 are down-core data.

We agree that the factor analysis description might have been unclear to readers who are not familiar with the method. A much more detailed presentation of the technique applied together with results of the analysis of the modern calibration data set is presented in the Supplementary Information. The downcore variability of the eight factor loadings is presented as well (Fig. S4).

Factors are listed in the order of the significance, i.e. to which modern factor assemblages the down core data correlate most.

All along the text, the use of the terms “warm” and “North Atlantic” is also confusing: Miettinen et al. (2015)- presenting the modern data-set- qualify the WGC as temperate, but the authors refers to warm waters from the North Atlantic (or Atlantic water inflow, Atlantic sourced waters...): then, why the factor corresponding to the North Atlantic Current is not also presented? More comments are listed below.

The characterisation of the West Greenland Current (WGC) as “warm”, which is used throughout the text is used to help the comparison of our study with other studies discussing Atlantic water influence. Such studies include Kuijpers et al., (2003), Jennings et al. (2006) and Dyke et al. (2014), who found incursion of warmer waters on the SE Greenland coast during the deglaciation and over the YD interval that they related to a strengthening of the IC.

The water masses associated with the WGC are definitely colder and fresher than the North Atlantic Current, but warmer than water masses typically found today north of Davis Strait. The modern distribution of the eight diatom assemblage factors based on the North-Atlantic diatom data set (Miettinen et al., 2015) (see Supplementary Information) shows the highest loadings of the East-West Greenland Current factor are distributed southwest and southeast of south Greenland and to the northeast of Iceland, and hence are geographically linked with a mix of the warm and saline Irminger Current and the cold and fresh WGC waters. The WGC is the source of Atlantic water to our study area.

The North Atlantic Current assemblage is dominated by diatom species that dwell in warmer waters (such as *Thalassiosira oestrupii*) (Andersen et al. 2004). These species are present in our record as trace species (at very low abundances) and thus the North Atlantic current assemblage has a very low factor loading (see Supplementary Information).

As mentioned previously all the factor loadings for the down core diatom data is presented in figure S4 in the Supplementary Information.

References:

- Andersen, C. et al. A highly unstable Holocene climate in the subpolar North Atlantic: evidence from diatoms. *Quat. Sci. Rev.* 23, 2155-2166 (2004).
- Dyke, L. M. et al. Evidence for the asynchronous retreat of large outlet glaciers in southeast Greenland at the end of the last glaciation. *Quat. Sci. Rev.* 99, 244-259 (2014).
- Jennings, A. et al. Freshwater forcing from the Greenland Ice Sheet during the Younger Dryas: evidence from southeastern Greenland shelf cores. *Quat. Sci. Rev.* 25, 282-298 (2006).
- Kuijpers, A. et al. Late Quaternary sedimentary processes and ocean circulation changes at the Southeast Greenland margin. *Mar. Geol.* 195, 109-129 (2003).
- Miettinen, A. et al. Exceptional ocean surface conditions on the SE Greenland shelf during the Medieval Climate Anomaly. *Paleoceanography* 30, 1657-1674 (2015).

Additional comments:

Lines 16-17: the YD was not the only interruption

Although climate indeed fluctuated during this transition, the YD was the only marked cold period, a fact we would like to emphasize.

Line 25: on which basis is an “amplified seasonality” interpreted?

The amplified climate seasonality during the YD was discussed in Broecker (2006), Denton et al. (2005), Björck et al., (2002) and Young et al., (2012). Evidence for a seasonal rather than perennial sea ice off the coast of north Norway (despite a very extensive winter sea ice cover in the NA) are found in Cabedo-Sanz et al., (2013). Our diatom data also supports a seasonal rather than a perennial ice cover during the cold YD period. Our reconstructed aSSTs are summer temperatures, and they are (apart from the transition into the Holocene) highest during the YD.

References:

Björck, S. et al. Anomalously mild Younger Dryas summer conditions in southern Greenland. *Geology* 30, 427-430 (2002).

Broecker, W.S. Abrupt climate change revisited. *Global and Planetary change* 54, 211-215 (2006).

Cabedo-Sanz, P., Belt, S.T., Knies, J.K., Husum, K. Identification of contrasting seasonal sea ice conditions during the Younger Dryas. *Quat. Sci. Rev.* 79, 74-86 (2013).

Denton, G. H. et al. The role of seasonality in abrupt climate change. *Quat. Sci. Rev.* 24, 1159-1182 (2005).

Young, N. E. et al. Glacier Extent During the Younger Dryas and 8.2-ka Event on Baffin Island, Arctic Canada. *Science* 337, 1330-1333 (2012).

Line 40-41: I guess the authors refer to the Arctic and there are more references on high resolution studies as ref. n°29.

Reference 29 (in the original submission) is indeed a high-resolution study, but the high resolution records in this paper are based on lake sediments, whereas we want to point out the need for marine proxy data. We have re-phased the sentence.

Line 64: what do you intend by large? Number and/or spatial distribution of the samples?

In this case we are referring to the number and spatial distribution of the samples, which is large compared to other existing diatom calibration dataset from the North Atlantic region (Jiang et al., 2001; Sha et al., 2014; Krawczyk et al., 2016) that are regionally very limited, which likely limits the temperature range of the data sets and causes problems with spatial autocorrelation.

References:

Jiang, H. et al. Diatom surface sediment assemblages around Iceland and their relationships to oceanic environmental variables. *Mar. micropal.* 41, 73-96 (2001).

Krawczyk, D. et al. Quantitative reconstruction of Holocene sea ice and sea surface temperature off West Greenland from the first regional diatom data set. *Paleoceanography*, 31, doi:10.1002/2016PA003003 (2016).

Sha, L. et al. A diatom-based sea-ice reconstruction for the Vaigat Strait (Disko Bugt, West Greenland) over the last 5000 yr. *Palaeogeography, Palaeoclimatology, Palaeoecology* 403, 66–79 (2014).

Line 79: it is high resolution indeed, but also a lot of noise and you had to smooth your record. Why not insisting instead on the quality of the sediments that suggests little bioturbation?

Smoothing was used to reveal the main trends in the record.

Line 95: reference for the description of the present day distribution of the assemblages? Miettinen et al 2015?

We added a reference.

Line 149: is there a fresh-water diatom record?

As stated previously, despite large melt events, surface salinities would still be relatively high (above 30 per mil, see e.g. Straneo et al., (2012) for current glacier-proximal locations). Hence one wouldn't find any fresh water diatoms due to melt events. In addition, true fresh water diatoms are usually only found in coastal settings, where rivers and streams enter the sea.

In this study we use the Marginal Ice Zone diatom assemblage to indicate meltwater influence. As stated in the manuscript (including references), these species favor the cold and fresher water layer produced by melting ice (sea ice and glaciers). These species inhabit both sea ice and open water, but typically form blooms in the melt layer overlying ambient sea water.

Line 189: there are more references for the mid-YD

We agree that there are several references for the mid-YD, yet we are not sure if Reviewer#3 means just mid-YD in general, or the climate shift occurring at 12.2 kyr BP during the mid-YD that we are referring to in the manuscript.

We have added one more reference to a study that points to a climate shift at 12.2 kyr BP (Gil et al., 2015).

References:

Gil, I.M., Keigwin, L.D. and Abrantes, F. The deglaciation over Laurentian Fan: History of diatoms, IRD, ice and fresh water. *Quat. Sci. Rev.* 129, 57-67 (2015).

Methods section, coring: I expected here the sampling strategy (not lines 231-232)

These lines have now been shifted into the methods, coring and sampling section (lines 257-259).

Lines 235-236: should cite Miettinen et al as it is a phrase copied from their manuscript!

This sentence has been rephrased.

Lines 248-249: axes listed in different order: explain why please. Why changing the denomination of Andersen et al group? Then, why considering them the equivalent.

Andersen et al. (2004) presents the modern surface assemblages based on a subset of the dataset used in this study. The order of listing the axes is not significant. In this study we listed the assemblages according which assemblages have highest loadings in the studied core.

The Sea Ice assemblage in Andersen et al. (2004) was re-named to Marginal Ice Zone assemblage in this study as these diatoms dwell in the cold, fresh surface water layer that is produced by melting glacier ice/sea ice. This name reflects more adequately the true (still sea ice related) ecology of these species.

References:

Andersen, C. et al. Nonuniform response of the major surface currents in the Nordic Seas to insolation forcing: Implications for the Holocene climate variability. *Paleoceanography* 19, PA2003 (2004).

Line 250: I don't understand at all this sentence

We have re-phased this sentence to make it comprehensible.

Lines 251-252: full name of the species are needed

We have added full names of the species.

Table 1: specify the fragments

Dated fragments are unidentified mollusc fragments, we have added this to the Table 1.

Mixed benthic (duplicate): I assumed they are foraminifera

Mixed benthics are foraminifera and we have added this to the Table 1.

Figure 2: It would be less confusing if all the description of the chronology would be in the method section

The legend does not include any information on the chronology that is not in the method section. It only mentions the number of the dated horizons and otherwise describes figure features.

Figure 3: no diatom abundances? It is sometimes referred that they are less abundant in ice/dissolved: would it provide more information?

We have now added diatom concentrations (see Supplementary Information, figure S2). They show very low concentrations before 15 cal kyr BP and similar, though rather variable, concentrations to current values in high Arctic surface sediments (in the order of 10^6 valves/g dry sediment (Polyakova 2003; Limoges et al. submitted to Journal of Geophysical Research – Biogeosciences).

References:

Limoges, A., Ribeiro, S., Weckström, K., Zamelczyk, K., Heikkilä, M., Andersen, T. J., Tallberg, P., Massé, G., Rysgaard, S., Nørgaard-Pedersen, N. and Seidenkrantz, M.-S. Linking the modern distribution of biogenic proxies in fjord sediments from High Arctic Greenland to sea-ice, primary production and Arctic-Atlantic Inflow. Submitted.
Polyakova, Y. I. Diatom assemblages in surface sediments of the Kara Sea (Siberian Arctic) and their relationships to oceanological conditions. In: R. Stein, K. Fahl, D.K. Fütterer, E.M. Galimova and O.V. Stepanets (Eds) Siberian River Run-off in the Kara Sea. Elsevier Science B.V. (2003).

This reference would have finally enriched the discussion:

Knutz, P. C. et al (2011), Multiple-stage deglacial retreat of the southern Greenland Ice Sheet linked with Irminger Current warm water transport, *Paleoceanography*, 26, PA3204, doi:10.1029/2010PA002053. Their cores are from nearby and even the YD SST are very limited (but compared to regional data), they discuss the inflow of warm Atlantic waters and the reliability of the warm signal. They also present an interesting IRD record in high-resolution and mention studies arguing for an increased inflow of Atlantic waters to the Nordic seas during the YD.

We have included this reference in the discussion.

Reviewers' comments:

Reviewer #2 (Remarks to the Author):

Review «Younger Dryas ice margin retreat triggered by warming of the ocean surface in central-eastern Baffin Bay» by Mimmi Oksman et al. submitted to Nature Communications.

The manuscript have been revised and has certainly benefitted greatly from this. I have only a few concerns left and some corrections.

The BBDC0 and BBDC1 are only shown in the supplement and only by age. It should be marked in the age model in Fig. 2 to give the exact location and correlation to the age model and lithology.

The temperature:

As stated in the previous review the entire sequence analyzed in core SL 170 is not cold at all and calculated temperature is $> 3\text{C}$ throughout. The new temperature calculations for the older part are not really cold either.

So instead of referring to a 'warm' Younger Dryas' it would be more accurate to state that no strong cooling in your studied time interval 14-10.2 ka was found. Calling the Younger Dryas interval 'warm' is still an overstatement in my opinion. Emphasizing in your manuscript that you see no cold event is equally interesting and more accurate to your actual results – and any change in the age model will not change that.

Therefore in Fig. 3: mark Younger Dryas boundaries – not 'warm' periods as it is now. The temperature scale is exaggerated on the y-axis – reduce this and instead indicate modern aSST (calculated by diatoms or measured) to emphasize that the entire study interval show warm temperatures and not cold – and some minor coolings, only.

Age model:

The error in the age model is not solved by experimenting with different deltaR (DR) values kept constant, because DR is not constant over time. A real experiment would be to vary the DR over time – for example by following the detailed development shown by Bondevik et al. 2006. Near normal DR in Bølling-Allerød and Holocene and high DR during the YD and gradual changes in transitions. The fact that the benthic-planktic 14C dates at 74-76 (Holocene) and 399-402 cm (c. Bølling/Allerød) indicates a small difference is not an argument that DR could not be much higher over the Younger Dryas interval. You could make the experiment that you keep DR in Holocene as modern, vary YD as you did (200, 400, 1000 years etc), and give Bølling-Allerød modern DR and a trial of DR c. 200 years (as the two planktic-benthic differences could suggest. This would give an entirely different record.

The BBDC0-layer has been found in other records and correlated to early Holocene and you can argue that this layer most likely is the layer that correlates with the previously published dates from the area (Jennings et al. 2014; Simon et al., 2014 a.o.). I think the authors should argue more clearly about the correlation of the BBDC-layers, and not just refer to Jackson et al. 2017 – and revise the text in the supplement lines 40-51. The age model for age of the BBDC-layers has the same problem with Delta-R, and is to me circular argumentation. The study of Simon et al., 2014 and 2012 used other means for the age model than just 14C.

Other points:

Line 50: "off" should be 'of'

Line 76: delete "therefore"

Line 83: "the vicinity" change to 'influence'

Line 91: cm and age are indicated oppositely

Lines 93-94: please also write the cm-interval so it is easier to find in the figure

Line 103: they are not "pronounced cold episodes", change to "slightly colder intervals" or something similar

Line 104: the decrease is not "marked", it is gradual over a long time interval, write the actual aSST decrease in centigrade and how long it takes

Line 106: "slightly warmer interval between ca. 11.4 and 11.1" - way too detailed for an insignificant change

Line 141: "SAT" - is that necessary?

Line 169: "surface warming" - change to 'warm sea surface conditions'

Line 172: not very clear. I suggest add to end of sentence: 'in that no strong cooling during the Younger Dryas is apparent in the SL170 record' - or something similar to better explain what is the "mismatch"

Line 176: "The reconstructed aSSTs show warmer conditions starting at ca. 13.4 kyr", change to: 'The reconstructed aSSTs show that warm conditions started before ca. 14 kyr'

Line 179: "associated" change to 'correlates'

Lines 179-180: you have one sharp peak of 'coarse' sediments (>63 micron), otherwise the sediment between 14 and 13.9 kyr is very fine grained. Give age of the peak.

Lines 186-187: "calving occurred in distinct episodes"-please be specific. I am confused what is meant by 'calving' - is that grain-size >63 micron? if so there is more calving before and after YD. Or are you relying on the sedimentation rates for these statements? Calculating the flux of the material > 63 micron would make the support for your statements stronger and more clear.

Lines 192-193: "improper uplift adjustment" - what is this? based on what? And choose different wording

Line 200: "East-West Greenland component" - I guess this comes from the diatom flora factor analysis - please be specific

Line 216: "cooled significantly", change to "cooled slightly"

Line 216: "following the termination of the YD" change to 'at the end of the YD'

Lines 217-220: confusing. When the YD ends it is Holocene. The cooler interval you are referring to belongs to the Holocene according to your correlation to the ice core.

Line 240: refer to lines 217-220 above: now it is called "transition to the Holocene". 10.9 and 10.8 kyr are in the Holocene as shown by the ice core.

Lines 240-241: "enhancement of the WGC" "high for this assemblage" - please specify and explain what is referred to here.

Line 248: "warmer ocean from ca. 13.4" change to 'warm ocean from 14'

Figures:

Fig. 2: mark (highlight) BBDC layers on age and cm axes

Fig. 3. I think the ice core should be taken off the figure - although you argue that an extra e.g. 400 reservoir correction does not change that the 'warm' interval belong to the YD - your record cannot be correlated directly to the ice core.

Reviewer #3 (Remarks to the Author):

The authors improved significantly their manuscript by considering the remarks of the reviewers and by complementing the supplementary information section. In particular, the evolution of the different diatom species validates the limited amplitude of the YD warming, much more than the addition pre-YD samples (not sufficient to be relevant). I also appreciated the details on the age model. However, the authors still use some shortcuts along the text which leads to inaccurate statements:

- The waters could be described as warmer during their YD warming (as they most of the time do), but not warm (lines 27, 271, 293 for example). Additionally, line 257, I guess they also assume "warmer summers and colder winters";
- The WGC is not Atlantic water: it is more accurate to write Atlantic sourced water or to use an equivalent expression (line 29 for example).

Considering that the manuscript is much more convincing, I would recommend it for publication if these minor inaccuracies are corrected.

Response to the reviewers' comments and suggestions on the revised manuscript

We thank reviewers for their very positive comments on our revision and much appreciate the opportunity to respond to the few remaining points raised. We have addressed all the comments; our responses and changes made to the manuscript are listed below and indicated in green in the manuscript. The original comments are in black and our responses in blue.

Reviewers' comments:

Reviewer #2 (Remarks to the Author):

Review «Younger Dryas ice margin retreat triggered by warming of the ocean surface in central-eastern Baffin Bay» by Mimmi Oksman et al. submitted to Nature Communications.

The manuscript have been revised and has certainly benefitted greatly from this. I have only a few concerns left and some corrections.

The BBDC0 and BBDC1 are only shown in the supplement and only by age. It should be marked in the age model in Fig. 2 to give the exact location and correlation to the age model and lithology.

BBDC0 and BBDC1 have been added to Figure 2.

The temperature:

As stated in the previous review the entire sequence analyzed in core SL 170 is not cold at all and calculated temperature is > 3C throughout. The new temperature calculations for the older part are not really cold either.

The reconstructed values for aSSTs depend on the range of aSST in the used modern calibration set. Our northern North-Atlantic training set has relatively few samples (8) with aSST <2°C so our reconstructed temperatures at low aSST are likely to be slightly higher than the real values would have been. However, our reconstructed temperatures are still well below the modern average summer (July) surface temperature of 5°C (Boertmann et al. 2003), which has been added to Figure 3.

So instead of referring to a 'warm' Younger Dryas' it would be more accurate to state that no strong cooling in your studied time interval 14-10.2 ka was found. Calling the Younger Dryas interval 'warm' is still an overstatement in my opinion. Emphasizing in your manuscript that you see no cold event is equally interesting and more accurate to your actual results – and any change in the age model will not change that.

We have combined the advice above and the comment of Reviewer#3 and have incorporated both the statement of "no strong cooling" at relevant places and refer to the YD throughout the manuscript as "warmer", not "warm".

Therefore in Fig. 3: mark Younger Dryas boundaries – not 'warm' periods as it is now. The temperature scale is exaggerated on the y-axis – reduce this and instead indicate modern aSST (calculated by diatoms or

measured) to emphasize that the entire study interval show warm temperatures and not cold – and some minor coolings, only.

We have removed the red shading indicating “warm periods” and have marked the boundaries of YD. We have further indicated the modern measured summer temperatures in Figure 3. We would, however, like to keep the temperature scale as it is, to underline the changes discussed in the manuscript. The temperature values are clearly indicated on the graph, so the reader is not in any way misled.

Age model:

The error in the age model is not solved by experimenting with different deltaR (DR) values kept constant, because DR is not constant over time. A real experiment would be to vary the DR over time – for example by following the detailed development shown by Bondevik et al. 2006. Near normal DR in Bølling-Allerød and Holocene and high DR during the YD and gradual changes in transitions. The fact that the benthic-planktic 14C dates at 74-76 (Holocene) and 399-402 cm (c. Bølling/Allerød) indicates a small difference is not an argument that DR could not be much higher over the Younger Dryas interval. You could make the experiment that you keep DR in Holocene as modern, vary YD as you did (200, 400, 1000 years etc), and give Bølling-Allerød modern DR and a trial of DR c. 200 years (as the two planktic-benthic differences could suggest. This would give an entirely different record.

We have done the experiment suggested by the reviewer, and included the results in the Supplementary Information (Fig. S2). The results suggest that the warmer sea surface temperatures still occur within the YD; even in the most extreme ($\Delta R=1000\pm 100$ years) experiment only the end of the warmer sequence is shifted into the early Holocene. The rapid advance and collapse of Jakobshavn Isbræ would still occur under warmer ocean conditions.

The BBDC0-layer has been found in other records and correlated to early Holocene and you can argue that this layer most likely is the layer that correlates with the previously published dates from the area (Jennings et al. 2014; Simon et al., 2014 a.o.). I think the authors should argue more clearly about the correlation of the BBDC-layers, and not just refer to Jackson et al. 2017 – and revise the text in the supplement lines 40-51. The age model for age of the BBDC-layers has the same problem with Delta-R, and is to me circular argumentation. The study of Simon et al., 2014 and 2012 used other means for the age model than just 14C.

We have referred twice to several previous studies on BBDC-events in the manuscript and once in the Supplementary Information (see below).

Lines 162-164: “The radiocarbon chronology is notably strengthened by the correlation of the detrital carbonate rich layers in core SL 170³⁹ with known Baffin Bay Detrital Carbonate events (BBDC)^{20,49-53}”.

Lines 283-285: “...value and by correlating the timing of detrital carbonate layers in the sediment³⁹ with the timing of the known Baffin Bay detrital carbonate events BBDC^{20,41,50,52,53} and BBDC1^{20,49,51,53}”.

Lines 46-49: “Based on our chosen chronology (with a ΔR of 140 ± 35 years), BBDC-events in core SL 170 have a timing of ca. 14.2-13.7 kyr BP and ca. 12.7-11 kyr BP and are found to be synchronous across the Baffin Bay¹⁰. The timing of these layers based on our chosen age model corresponds well with previously identified BBDC1¹⁶⁻¹⁹ and BBDC0^{3,18-21} events”.

The reviewer is completely right about the ΔR issue regarding the defining of the timing of BBDC events in studies using only radiocarbon dating (i.e. the majority of studies on BBDC events). Simon et al. 2012 and 2014 have in addition to radiocarbon dating created a paleomagnetic-based chronology, and have arrived at largely the same ages for BBDC0 and BBDC1 as the other studies. However, uncertainties also exist in

paleomagnetic-based chronologies, due to the ambiguity in tuning noisy curves to a target curve that, in itself, includes dating uncertainty.

Other points:

Line 50: "off" should be 'of' This has been changed.

Line 76: delete "therefore" Deleted.

Line 83: "the vicinity" change to 'influence' This has been changed.

Line 91: cm and age are indicated oppositely This has been changed.

Lines 93-94: please also write the cm-interval so it is easier to find in the figure Cm-intervals have been added.

Line 103: they are not "pronounced cold episodes", change to "slightly colder intervals" or something similar The word "pronounced" is removed. The sentence now reads: "... and is punctuated by colder episodes..."

Line 104: the decrease is not "marked", it is gradual over a long time interval, write the actual aSST decrease in centigrade and how long it takes Word "marked" has been changed to "notable gradual" and centigrade and time interval are given.

Line 106: "slightly warmer interval between ca. 11.4 and 11.1" - way too detailed for an insignificant change This sentence has been removed.

Line 141: "SAT" – is that necessary? This has been removed.

Line 169: "surface warming" – change to 'warm sea surface conditions' Done.

Line 172: not very clear. I suggest add to end of sentence: 'in that no strong cooling during the Younger Dryas is apparent in the SL170 record' – or something similar to better explain what is the "mismatch" "...which both, unlike the SL 170 record, point to strong cooling during the Younger Dryas" was added to the text.

Line 176: "The reconstructed aSSTs show warmer conditions starting at ca. 13.4 kyr", change to: 'The reconstructed aSSTs show that warm conditions started before ca. 14 kyr' We have changed the sentence to "Instead of recording a cooling of the ocean surface during the YD cold period, the reconstructed aSSTs show warmer conditions (compared to a SSTs before and after YD) starting at ca. 13.4 kyr BP and continuing to the end of the YD (Fig. 3a).

Line 179: "associated" change to 'correlates' Changed.

Lines 179-180: you have one sharp peak of 'coarse' sediments (>63 micron), otherwise the sediment between 14 and 13.9 kyr is very fine grained. Give age of the peak. Age added.

Lines 186-187: "calving occurred in distinct episodes"-please be specific. I am confused what is meant by 'calving' – is that grain-size >63 micron? if so there is more calving before and after YD. Or are you relying

on the sedimentation rates for these statements? Calculating the flux of the material > 63 micron would make the support for your statements stronger and more clear.

We have rewritten lines 183-188. Our statements are based on the high sedimentations rates, the decreasing amount of finer-grained sediments indicating increased distance between the core site and the glacier, and on the high variability in the diatom data between 12.5 and 12.1 kyr BP, which strongly points towards calving and associated melting of icebergs occurring in episodes.

Lines 192-193: "improper uplift adjustment" – what is this? based on what? And choose different wording
The sentence has been reworded.

Line 200: East-West Greenland component" – I guess this comes from the diatom flora factor analysis – please be specific We have added "Fig. 3d"

Line 216: "cooled significantly", change to "cooled slightly" The sentence has been reworded.

Line 216: "following the termination of the YD" change to 'at the end of the YD' See above.

Lines 217-220: confusing. When the YD ends it is Holocene. The cooler interval you are referring to belongs to the Holocene according to your correlation to the ice core. Word "towards" has been replaced with word "in".

Line 240: refer to lines 217-220 above: now it is called "transition to the Holocene". 10.9 and 10.8 kyr are in the Holocene as shown by the ice core. Transition to the Holocene has been removed.

Lines 240-241: "enhancement of the WGC" "high for this assemblage" – please specify and explain what is referred to here. Reference to Table S2 has been added.

Line 248: "warmer ocean from ca. 13.4" change to 'warm ocean from 14' The sentence has been reworded.

Figures:

Fig. 2: mark (highlight) BBDC layers on age and cm axes BBDC layers have been added to Figure 2.

Fig. 3. I think the ice core should be taken off the figure – although you argue that an extra e.g. 400 reservoir correction does not change that the 'warm' interval belong to the YD – your record cannot be correlated directly to the ice core.

We are also correlating our reconstruction with other previously published data (the time-distance diagram of Jakobshavn Isbræ extent modified from Ó Cofaigh et al. 2013 and the AMOC rate presented as sedimentary $^{231}\text{Pa}/^{230}\text{Th}$ from the subtropical North Atlantic Ocean (McManus et al. 2004). We think these additional data, including the stable oxygen isotope $\delta^{18}\text{O}$ record from the NGRIP Greenland Ice Core, provide an essential framework for our own data and interpretations. However, we completely understand the reviewer's concerns and have added a sentence to the figure caption referring to the dating uncertainties and the ΔR experiments in the Supplementary Information.

Reviewer #3 (Remarks to the Author):

The authors improved significantly their manuscript by considering the remarks of the reviewers and by complementing the supplementary information section. In particular, the evolution of the different diatom species validates the limited amplitude of the YD warming, much more than the addition pre-YD samples (not sufficient to be relevant). I also appreciated the details on the age model. However, the authors still use some shortcuts along the text which leads to inaccurate statements:

- The waters could be described as warmer during their YD warming (as they most of the time do), but not warm (lines 27, 271, 293 for example). Additionally, line 257, I guess they also assume “warmer summers and colder winters”; These have been changed accordingly.

- The WGC is not Atlantic water: it is more accurate to write Atlantic sourced water or to use an equivalent expression (line 29 for example). Atlantic water has been changed to Atlantic-sourced water.

Considering that the manuscript is much more convincing, I would recommend it for publication if these minor inaccuracies are corrected.

REVIEWERS' COMMENTS:

Reviewer #2 (Remarks to the Author):

Review «Younger Dryas ice margin retreat triggered by warming of the ocean surface in central-eastern Baffin Bay» by Mimmi Oksman et al. submitted to Nature Communications no. NCOMMS-16-29394A

I greatly appreciate the second revision of the manuscript, the reply to the comments. The manuscript is now acceptable.